   

# Modulating tumor immunity using advanced microbiome therapeutics producing an indole metabolite

Troels Holger Vaaben[1], Ditte Olsen Lützhøft[1], Andreas Koulouktsis [ID][1], Ida Melisa Dawoodi[2], Camilla Stavnsbjerg[2], Lasse Kvich[3], Ismail Gögenur[3], Ruben Vazquez-Uribe [ID][1,4] & Morten Otto Alexander Sommer[1]✉

## Abstract

**The gut microbiome has emerged as a key player in modulating immune responses against cancer, suggesting that microbial interventions can enhance treatment outcomes. Indole metabolites produced by probiotic bacteria activate the aryl hydrocarbon receptor (AhR), a transcription factor important for immune cell regulation. Cancer patients with high plasma concentrations of these metabolites have shown improved survival. Building on these findings, we have engineered *Escherichia coli* Nissle 1917 to produce the AhR agonist indole-3-acetic acid. Delivery of indole-3-acetic acid by tumor-colonizing bacteria changes the tumor micro-environment in a murine model, significantly increasing levels of CXCL9 and IFN-γ and elevating tumor-infiltrating T-cell abundance and activation. Treatment with our engineered strain inhibits tumor growth, improves survival in syngeneic tumor models, and leads to long-lasting immunity in a tumor rechallenge experiment. Further investigation indicates that this immune modulation is driven by the direct activation of AhR by indole-3-acetic acid, leading to differential cytokine expression and a shift in immune cell composition within the tumor. This study highlights the importance of microbial metabolites in immune modulation and supports exploring microbiome-based therapies in oncology.**

**Keywords** Microbiome Therapeutics; Indole-3-Acetic Acid; Aryl Hydrocarbon Receptor; Tumor Immunomodulation; Synthetic Biology
**Subject Categories** Cancer; Immunology; Microbiology, Virology & Host Pathogen Interaction

## Introduction

Cancer poses a significant global health challenge due to its high mortality rates and substantial financial burden. Over the past two decades, therapies that direct the immune response against cancer targets have proven their utility, with immune checkpoint inhibitors dramatically improving the survival of patients with multiple cancer types, particularly in locally advanced and metastatic settings (Morad et al, 2021). Despite this success, developing effective strategies using immune checkpoint inhibitors has plateaued (Huang and Zappasodi, 2022) and remains limited to a subset of cancers with high tumor mutational burden. Novel approaches that activate the immune system against cancer could stimulate progress in new immunotherapy strategies.

Cancer cells reprogram their metabolism to reshape the tumor microenvironment thus sustaining continuous growth and avoiding immune destruction (Hanahan and Weinberg, 2011). However, differences in response to therapy are dictated not only by cancer-intrinsic properties but also extrinsic factors, including the patient's diet (Wang and Geng, 2023), exercise habits (van Rooijen et al, 2018), and microbiome. Specifically, the gut microbiome has demonstrated a causal role in the therapeutic response to immune checkpoint inhibitors and cancer outcomes in preclinical models (Routy et al, 2018) and patient cohorts (Heshiki et al, 2020; Gunjur et al, 2024). These findings spurred interest in developing interventions, including prebiotics, probiotics, and fecal-matter transplantation, to modulate the microbiome and improve the efficacy of existing treatments (Li et al, 2022).

The aryl hydrocarbon receptor (AhR), a key transcription factor in involved in cell homeostasis and immune cell responses (Rothhammer and Quintana, 2019), plays a central role in the interaction between microbes and immune cells, enabling gut microbial metabolites to exert an impact on immune responses. AhR is a promiscuous receptor capable of interacting with a broad spectrum of ligands (Larigot et al, 2022; Giani Tagliabue et al, 2019). This interaction can result in pro-inflammatory and

[1]Novo Nordisk Foundation Center for Biosustainability, Technical University of Denmark, Kgs. Lyngby DK2800, Denmark. [2]Department of Clinical Physiology and Nuclear Medicine & Cluster for Molecular Imaging, Copenhagen University Hospital—Rigshospitalet & Department of Biomedical Sciences, University of Copenhagen, Copenhagen 2200, Denmark. [3]Center for Surgical Science, Department of Surgery, Zealand University Hospital, Køge, Region Zealand 4690, Denmark. [4]Center for Microbiology, Vlaams Instituut voor Biotechnologie, Leuven, Belgium. ✉E-mail: msom@bio.dtu.dk

anti-inflammatory immune cell phenotypes, depending on the ligand and specific site of action (Bourner et al, 2022). This dichotomous nature has led to the AhR being considered both an oncogene and a tumor-suppressor across different types of cancer (Elson and Kolluri, 2023; Murray et al, 2014). Recent studies have underlined the positive impact of microbially produced indole derivatives, products of tryptophan metabolism, in inhibiting cancer growth through AhR-dependent (Bender et al, 2023) and independent mechanisms (Tintelnot et al, 2023; Jia et al, 2024).

Microorganisms do not only influence immune system homeostasis from the gut, as intratumoral microbiota has been demonstrated in many solid cancers (Hamada et al, 2023; Ma et al, 2020; Nejman et al, 2020; Geller et al, 2017). The modulation of the tumor microenvironment by the intratumoral microbiome is intricately linked to the microbial composition, the molecular crosstalk between these microorganisms and neoplastic cells, and the status of malignancy. The tumor microbiome can potentially regulate neoplastic cell biology and host immune reactions, influencing treatment response (Liu et al, 2023).

The presence of an intratumoral microbiome and the ability of certain bacteria to persist and proliferate within tumors has spurred an interest in harnessing bacteria as self-renewing vehicles for local delivery of payloads in solid tumors (Gurbatri et al, 2020; Leventhal et al, 2020; Redenti et al, 2024; Tumas et al, 2023). *Escherichia coli* Nissle 1917 (EcN) has been extensively investigated as a microbial vector for drug delivery and modulation of the tumor microenvironment across preclinical (Chowdhury et al, 2019; Stritzker et al, 2007; Tumas et al, 2023; Xie et al, 2017; Zhang et al, 2012) and clinical cancer research (Luke et al, 2023). Current efforts in this field can broadly be categorized into leveraging the natural ability of certain bacteria to stimulate the immune system and using bacteria as a delivery method for existing modalities. Using live bacteria as an alternative delivery method aims to improve treatment efficacy via local delivery and avoid toxicity associated with systemic administration of payloads such as immune checkpoint inhibitors (Gurbatri et al, 2020), IL-2 (Tumas et al, 2023) or cytotoxic agents (Din et al, 2016).

In this work, we build on recent discoveries that certain bacterial species from the gut microbiome can migrate to distal tumors in mice and produce compounds that activate the AhR (Bender et al, 2023), thereby stimulating an immune response against cancer. Our study exploits recent advances in microbiome synthetic biology to harness and amplify this microbial signature by genetically engineering EcN to produce the AhR agonist indole-3-acetic acid (IAA). We used the therapeutic approach of direct injection of EcN into solid murine tumors to investigate its potential as a novel advanced microbiome therapeutic (AMT) for modulating the tumor microenvironment.

## Results and discussion

### Engineering probiotic *E. coli* Nissle 1917 to produce IAA

For recombinant production of IAA in EcN, a three-step biosynthetic pathway was cloned on the cryptic pMUT1 plasmid under the control of a constitutive promoter (MS6) designed for use in vivo (Armetta et al, 2021) and two strong predicted ribosomal binding site (RBS) variants (Bonde et al, 2016). In the generated operon, the aspartate aminotransferase *aspC* gene from *E. coli* K-12 (UniProtKB entry P00509) conferred the ability to convert tryptophan to indole-3-pyruvate, followed by the decarboxylase *ipdC* gene from *Enterobacter cloacae* (UniProtKB entry P23234) and the dehydrogenase gene *iad1* from *Ustilago maydis* (UniProtKB entry P09317) to allow, respectively, production of the intermediate indole-3-acetaldehyde and the final product IAA, as previously described (Romasi and Lee, 2013) (Fig. 1A). All genes were codon-optimized for expression in *E. coli* (Fig. EV1). A *tnaA* knockout (Δ*tnaA*) EcN was generated to abolish indole production from the engineered biotherapeutic and was used as the background strain for all experiments. The tnaA knockout was confirmed by PCR and gel electrophoresis (Fig. EV2A), followed by Sanger sequencing. We evaluated three distinct combinations of RBSs, in conjunction with the constitutive MS6 promoter, within the pMUT1 plasmid (Fig. 1A). Quantitative analysis of tryptophan-derived metabolites was conducted on the spent medium from the engineered strains and the control strain harboring the empty pMUT1 (EcN^Ctrl) vector (Fig. 1B; Table 1) following 24 hours of fermentation. Notably, one of the strains (hereafter referred to as EcN^IAA) exhibited a 150-fold increase in IAA production and was used in subsequent experiments. We evaluated the effect of the Δ*tnaA* knockout and the introduction of the expression cassette containing aspC, ipdC, and iad1 on the growth performance of the strains and observed no notable differences, indicating that the simultaneous expression of the three payloads did not significantly impact bacterial growth (Fig. EV2B,C). These results demonstrate that we have successfully engineered a strain capable of secreting high quantities of IAA without compromising bacterial fitness. The mechanism by which IAA is exported by the bacteria remains unknown but may occur through native efflux pumps or through passive diffusion across the lipid membrane (Piñero-Fernandez et al, 2011).

Tryptophan-derived metabolites have the potential to act as AhR agonists. Accordingly, we examined the ability of IAA to activate AhR using a mammalian AhR reporter cell line (Fig. 1C). We observed a dose-dependent increase in signal upon exposure to IAA (Fig. 1D) confirming the ability of this metabolite to activate AhR signaling. Next, we investigated the ability of EcN^IAA to activate the AhR (Fig. 1E). A dose-dependent elevation in receptor activation was observed with medium from EcN^IAA compared to medium from the control strain and the media control. The discrepancy in AhR activity between pure IAA exposure and spent medium treatment may be attributed to the presence of other substances in the spent medium, such as LPS, which could induce stress responses. In addition, we observed that using more than 20% spent medium in the assay caused a color change, likely due to a shift in pH, which may have further influenced the AhR response. We did not detect elevated levels of other tryptophan-derived metabolites (Fig. 1B) from EcN^IAA compared to the control, indicating that AhR activation by the engineered strain was attributed to the presence of IAA. We tested the ability of the wild-type strain and the Δ*tnaA* knockout to activate AhR and found no difference (Fig. EV2D), further supporting the conclusion that AhR activation by our engineered strains is driven by the production of IAA. The control strain's ability to activate the AhR, despite the targeted disruption of indole production, implies that EcN may synthesize alternative activating compounds, reflecting

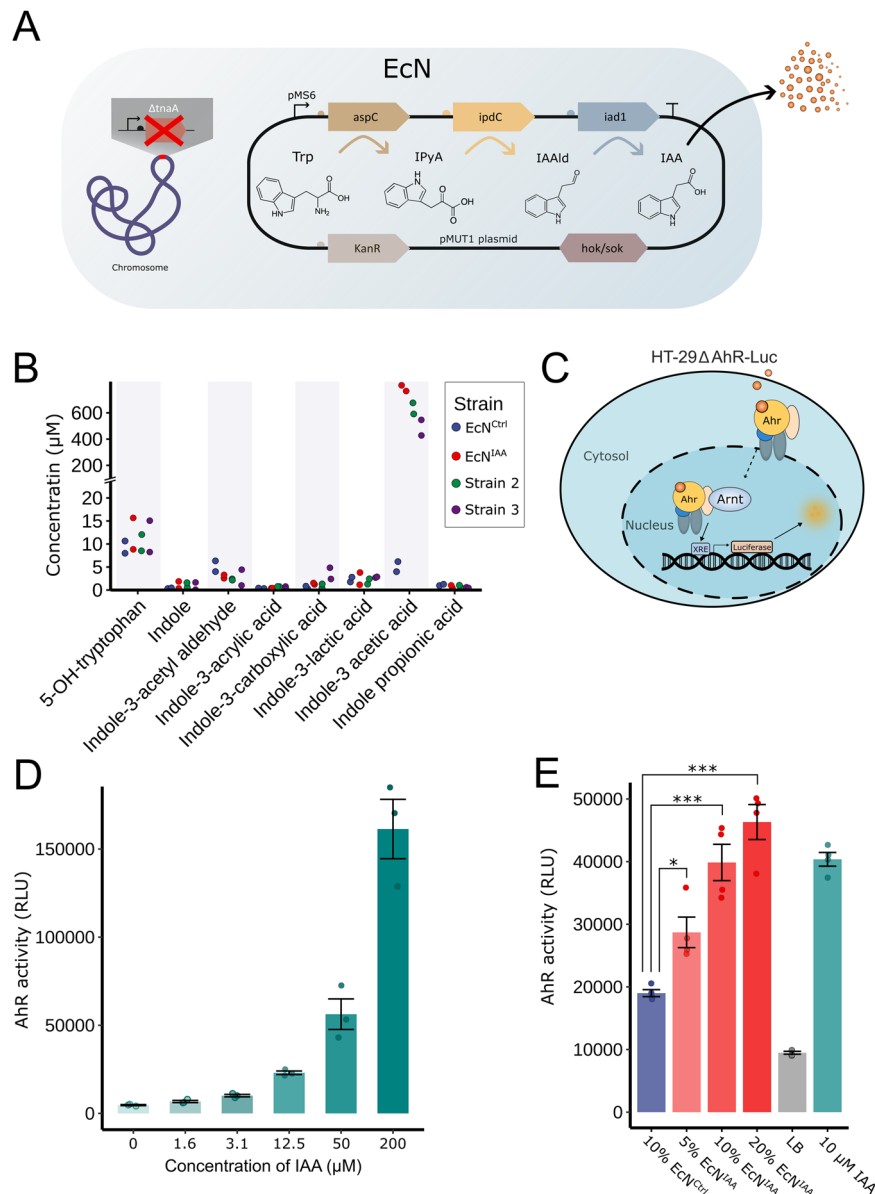

**Figure 1. Engineered *E. coli* Nissle 1917 producing IAA activates AhR.**

(A) Biosynthetic pathway introduced into *E. coli* Nissle 1917 (EcN) to produce indole-3-acetic acid (IAA). The endogenous pathway for indole synthesis was abolished by knocking out tnaA-encoded tryptophanase in the genome. (B) Quantification of tryptophan-derived metabolites in spent medium from EcN with an empty expression cassette plasmid (EcN$^{Ctrl}$) and three strains with different ribosome binding site combinations ($n = 2$ biological replicates) using liquid chromatography–high-resolution mass spectrometry. (C) Schematic of AhR reporter assay system used to measure AhR activity in the presence of bacterial supernatant or IAA. (D) Dose–response curve showing AhR activity as relative luminescent units (RLU) derived from luciferase-expressing AhR reporter cells following stimulation with increasing concentrations of IAA ($n = 3$ biological replicates). (E) AhR activity as RLU derived from AhR reporter cells stimulated with 5, 10, or 20% supernatant of either EcN$^{IAA}$, EcN$^{Ctrl}$, lysogeny broth (LB), or media with 10 μM IAA spike-in for 48 h ($n = 4$ biological replicates). Data are mean ± SEM, with overall differences in means determined by ANOVA, and post hoc comparison comparisons between groups using Tukey's honest significant difference test. *$P \leq 0.05$, ***$P \leq 0.001$. (E: 5%-EcN$^{IAA}$–EcN$^{Ctrl}$ $P = 0.0118$, 10% EcN$^{IAA}$–EcN$^{Ctrl}$ $P = 3.7e-06$, 20% EcN$^{IAA}$ –EcN$^{Ctrl}$ $P = 6.6e-08$. Source data are available online for this figure.

the capacity of AhR to respond to a diverse array of ligands (Larigot et al, 2022; Giani Tagliabue et al, 2019). In addition, the rich LB medium itself is likely to contribute to baseline AhR activation due to the presence of various small molecules.

These data demonstrate that our engineered EcN$^{IAA}$ can produce and secrete 780.14 (± 13.96) μM concentrations of the microbial metabolite IAA with a robust ability to activate the AhR.

## Microbial delivery of IAA mediates anti-tumor immunity

A recent study revealed that the translocation of indole-3-carbaldehyde (I3A) producing gut microbes into tumors in mice had a significant effect on the tumor microenvironment. Further, their findings implied that I3A could influence overall survival in patients with advanced melanoma undergoing immune checkpoint inhibitor therapy through

**Table 1. Targeted metabolomics for tryptophan-derived compounds from three strains of EcN engineered to produce IAA ($n = 2$ biological replicates, mean ± SD).**

| | MS6-RBS4-aspC-RBS4-ipdC-RBS3-iad1 (EcN[IAA]) | MS6-RBS3-aspC-RBS4-ipdC-RBS3-iad1 (Strain 2) | MS6-RBS4-aspC-RBS3-ipdC-RBS4-iad1 (Strain 3) | EcN[Ctrl] |
|---|---|---|---|---|
| Indole-3 acetic acid (MW 175.18) | 780.14 (± 13.96) | 633.73 (± 42.37) | 487.19 (± 59.28) | 5.08 (±1.09) |
| Indole-3-acetyl aldehyde (MW 145.16) | 2.92 (± 0.36) | 2.42 (± 0.13) | 2.72 (±1.71) | 5.17 (±1.16) |
| Indole-3-acrylic acid (MW 187.19) | 0.32 (± 0.08) | 0.58 (± 0.02) | 0.52 (± 0.25) | 0.43 (± 0.051) |
| Indole-3-lactic acid (MW 205.21) | 2.48 (±1.33) | 1.89 (± 0.56) | 2.78 (± 0.09) | 2.28 (± 0.51) |
| Indole propionic acid (MW 189.21) | 0.71 (± 0.3) | 0.81 (± 0.23) | 0.53 (± 0.08) | 1.11 (± 0.15) |
| Indole-3-carboxylic acid (MW 161.16) | 1.38 (± 0.16) | 0.88 (± 0.45) | 3.63 (±1.22) | 0.56 (± 0.29) |
| Indole-3-pyruvate (MW 203.19) | NA | NA | NA | NA |
| Indole (MW 117.15) | 1.12 (± 0.76) | 1.10 (± 0.50) | 0.87 (± 0.79) | 0.40 (± 0.07) |
| 5-OH-tryptophan (MW 220.22) | 12.25 (± 3.41) | 10.28 (±1.77) | 11.66 (± 3.41) | 9.32 (±1.33) |

*EcN Escherichia coli*, Nissle 1917, *IAA* indole-3 acetic acid, *Ctrl* control, *NA* not applicable.

an AhR-dependent mechanism (Bender et al, 2023). In light of these findings, including the translocation of the microbes from the gut to the tumor in mice, we investigated whether EcN[IAA] could modulate the tumor microenvironment by producing the microbial metabolite and AhR agonist IAA directly in tumors. Syngeneic murine tumor models with a complete immune landscape allow a translational approach to study the efficacy of new immunotherapies. CT26 cells are a commonly used murine colorectal cancer (CRC) model, with clinically relevant CRC mutations (Castle et al, 2014) and high infiltration of immune cells. BALB/c mice with 75–200 mm³ engrafted tumors were randomized into two groups receiving a single intratumoral injection of $10^7$ colony-forming units (CFU) of EcN[IAA] or EcN[Ctrl] (Fig. 2A).

A significant reduction in tumor growth was observed in mice treated with EcN[IAA] compared to those receiving the control strain (Fig. 2B). Treatment with EcN[IAA] significantly improved the overall survival in CT26 tumor-bearing mice, which were euthanized at predefined humane endpoints (Fig. 2C). The presence of viable bacteria in tumors was determined by plating serial dilutions of tumor homogenates on plates containing kanamycin. All animals harbored viable bacteria in tumors at the time of euthanization (Fig. 2C), confirming both the ability of EcN to persist within tumors over time and the long-term stability of the pMUT1-based plasmid, even in the absence of antibiotic selective pressure. The administration of bacteria was well tolerated in all animals, with no significant changes in bodyweight in either group (Fig. 2D). We also investigated potential off-target colonization and found no viable bacteria in livers, the organ most associated with off-target colonization (Stritzker et al, 2007; Tumas et al, 2023), at the time of euthanization (Fig. 2E). Spleen enlargement is commonly observed in transplantable tumor models (Bronte and Pittet, 2013; Li et al, 2016), and spleen volume is used as a surrogate biomarker for myeloid-derived suppressor cell accumulation and treatment response in several human cancers (Niogret et al, 2020; Bian et al, 2023). Tumor-bearing mice treated with EcN[IAA] had a decrease in spleen weight, although this difference was not statistically significant (Fig. 2F).

Flow cytometric analysis of frozen tumor tissues revealed that EcN[IAA] treatment shifted the immune cell composition, with a greater percentage of CD45[+] leukocytes expressing CD3[+], indicative of an enriched presence of T cells in animals treated with EcN[IAA] (Fig. 2G). Due to reduced cell viability resulting from the cryopreservation, cell counts within CD4[+] and CD8[+] populations

were insufficient (<500 cells) for meaningful analysis of downstream subsets.

Our findings indicate that treatment with EcN is well tolerated, consistently leading to successful tumor colonization in all the tested animals and with no observed off-target colonization. Notably, EcN[IAA] significantly reduced tumor volume, enhancing survival in the experimental model.

## EcN[IAA] increased the abundance of tumor-infiltrating lymphocytes

Elevated I3A concentrations in tumors have been shown to stimulate adaptive immunity (Bender et al, 2023), suggesting that IAA could foster similar results. To assess whether EcN[IAA] stimulated adaptive immune cells, three animals with size-matched tumors from the EcN[IAA] and the EcN[Ctrl] groups underwent histological analysis using immunohistochemistry for markers of CD4[+], CD8[+], FOXP3[+], and granzyme B (GrzB)[+] on serial sections from the same tumor (Fig. 3A). We also evaluated the effect on myeloid cells using CD68 as a marker for macrophages and monocytes, and ELA2 (neutrophil elastase) as a marker for neutrophils. High-resolution whole-slide images of specimens were captured, and QuPath software (Bankhead et al, 2017) was used for detailed segmentation and counting of each specimen's total and marker-positive cells (Fig. 3B). A significant increase in the abundance of CD4[+] and CD8[+] cells was observed in animals treated with EcN[IAA] (Fig. 3C). Interestingly, as the percentage of FOXP3[+] regulatory T cells remained unchanged between the groups, the data suggest that EcN[IAA] treatment increases the ratio of CD4[+] helper T cells to regulatory T cells, favoring a more pro-inflammatory adaptive immune response. In addition, a non-significant trend toward a higher abundance of granzyme B-positive cells ($P = 0.069$) was observed in the treatment group, indicative of a more cytotoxic milieu. We observed no significant differences in the counts of CD68[+] macrophages/monocytes or ELA2[+] neutrophils between the groups, suggesting that EcN[IAA] treatment primarily influences adaptive immune cells rather than myeloid cell populations.

The presence of EcN within the tumors was also confirmed using fluorescence in situ hybridization to investigate the spatial distribution of bacteria in the tumor microenvironment. We observed that EcN primarily colonized tissue adjacent to necrotic

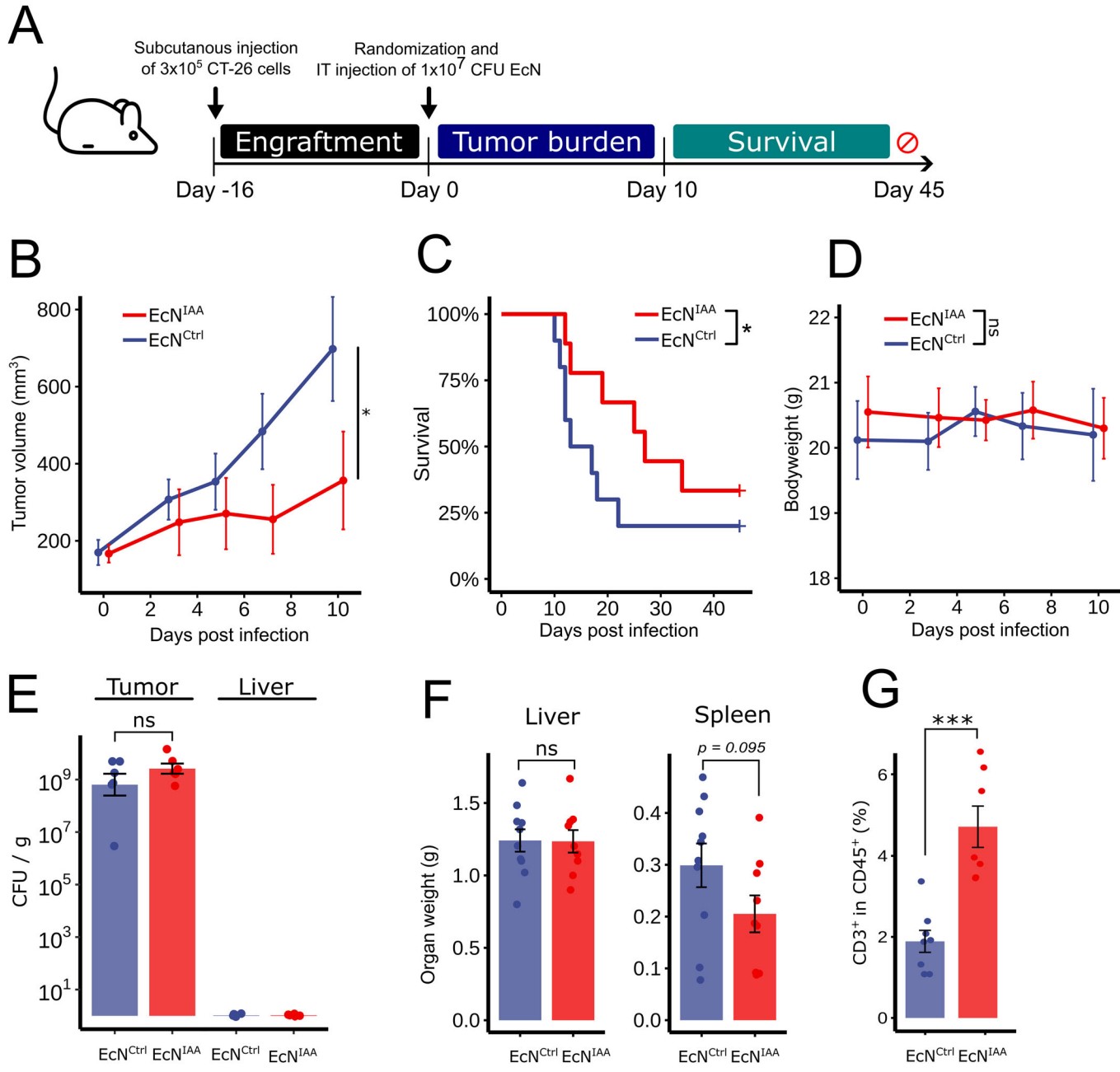

**Figure 2. EcN-mediated delivery of IAA reduces tumor burden in a colorectal syngeneic tumor model.**

(A) Schematic representation of experiments using the CT26 model. Seven-week-old female BALB/c mice were subcutaneously implanted with CT26 cells on their right flank ($n \geq 9$ per group) that grew to a 75–200-mm³ tumor before mice were randomized and intratumorally injected with $10^7$ colony-forming units (CFU) EcN$^{Ctrl}$ or EcN$^{IAA}$ in 20 µL PBS (Day 0). Tumor size was measured three times per week using a caliper, and survival was followed until day 45, after which all remaining animals were tumor-free. (B) Mean tumor volume trajectories. (C) Kaplan–Meier survival analysis censored at 45 days after three consecutive measurements with no tumors. (D) Bodyweight trajectories. (E) Bacterial colonization in tumors and livers, quantified as CFU per gram of organ. Individual points represent the mean of each animal, calculated from 4 technical replicates. (F) Liver and spleen weights. (G) Percentage of CD3$^+$ cells from tumors of animals treated with EcN$^{Ctrl}$ or EcN$^{IAA}$. Data represent the frequency of CD3$^+$ cells as a percentage of the parent population for each group. For each animal, half the tumor was sectioned and preserved by snap-freezing in liquid nitrogen. Samples were simultaneously thawed for flow cytometry, employing a T-cell-specific panel (see Fig. EV4 for gating strategy). Data are mean ± SEM. Statistical significance was determined using Wilcoxon rank-sum at the final observation timepoint preceding the first euthanasia according to humane endpoints (B), log-rank test (C), and two-tailed Student's $t$ test (E, F). *$P \leq 0.05$, ***$P \leq 0.001$. (B: $P = 0.04$, C: $P = 0.014$, G: $P = 7.896e-4$). Source data are available online for this figure.

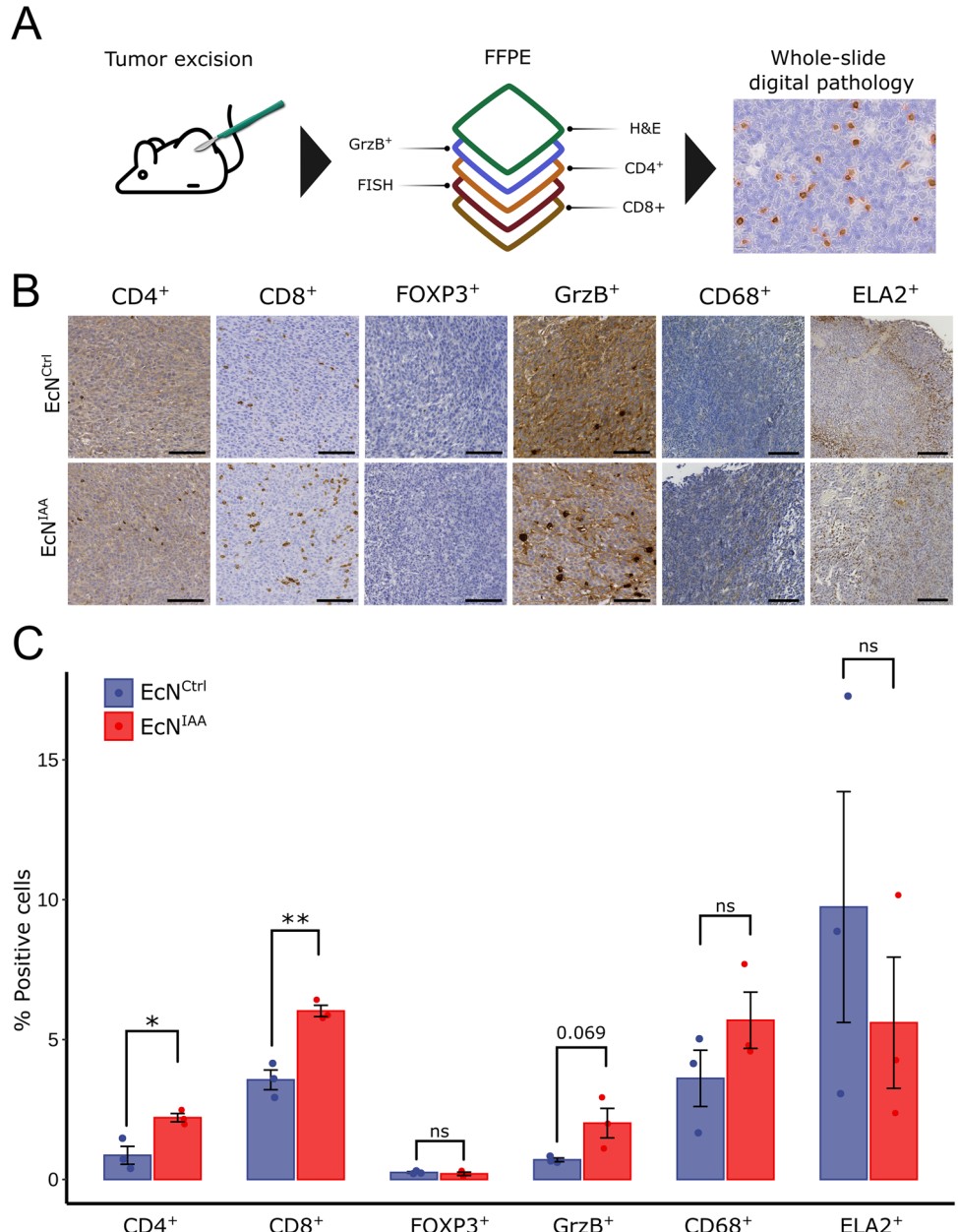

**Figure 3. EcN^IAA increases the abundance of tumor-infiltrating lymphocytes.**

(A) Workflow schematic for histological analysis. Tumors of comparable size from EcN^IAA (*n* = 3) and EcN^Ctrl (*n* = 3) animals were analyzed as formalin-fixed paraffin-embedded (FFPE) sections using H&E staining, fluorescence in situ hybridization (FISH), and immunohistochemical detection of CD4+, CD8+, FOXP3+, granzyme B+, CD68+, and ELA2+ cells. (B) Representative immunohistochemistry images compare animals treated with EcN^Ctrl (top) and EcN^IAA (bottom), scale bar = 100 μm. (C) Quantitative analysis of marker-positive cells on whole-slide images from tumor sections of animals treated with EcN^Ctrl (*n* = 3 biological replicates) and EcN^IAA (*n* = 3 biological replicates). Data are mean ± SEM. Statistical significance was determined with ANOVA and post hoc comparison comparisons between groups using Tukey's honest significant difference test (C). *$P \leq 0.05$, **$P \leq 0.01$. (C: CD4+ $P = 0.0189$, CD8+ $P = 0.00375$). Source data are available online for this figure.

regions, aligning with previous research describing preferential bacterial colonization in hypoxic tumor areas (Stritzker et al, 2007; Kvich et al, 2024). We observed no noticeable differences in colonization patterns between EcN^IAA and EcN^Ctrl (Fig. EV3).

The observed elevation in CD4+ and CD8+ T-cell counts in EcN^IAA-treated tumors indicates a marked stimulation of the adaptive immune system. This enhanced immune response could be driven by increased recruitment or proliferation of T cells within

the tumor microenvironment, highlighting the immunomodulatory potential of EcN^IAA treatment.

## EcN^IAA drives local and global changes in cytokine expression

To elucidate the mechanisms driving the observed immunological shift, we measured indole derivatives in tumors (Fig. 4A) and

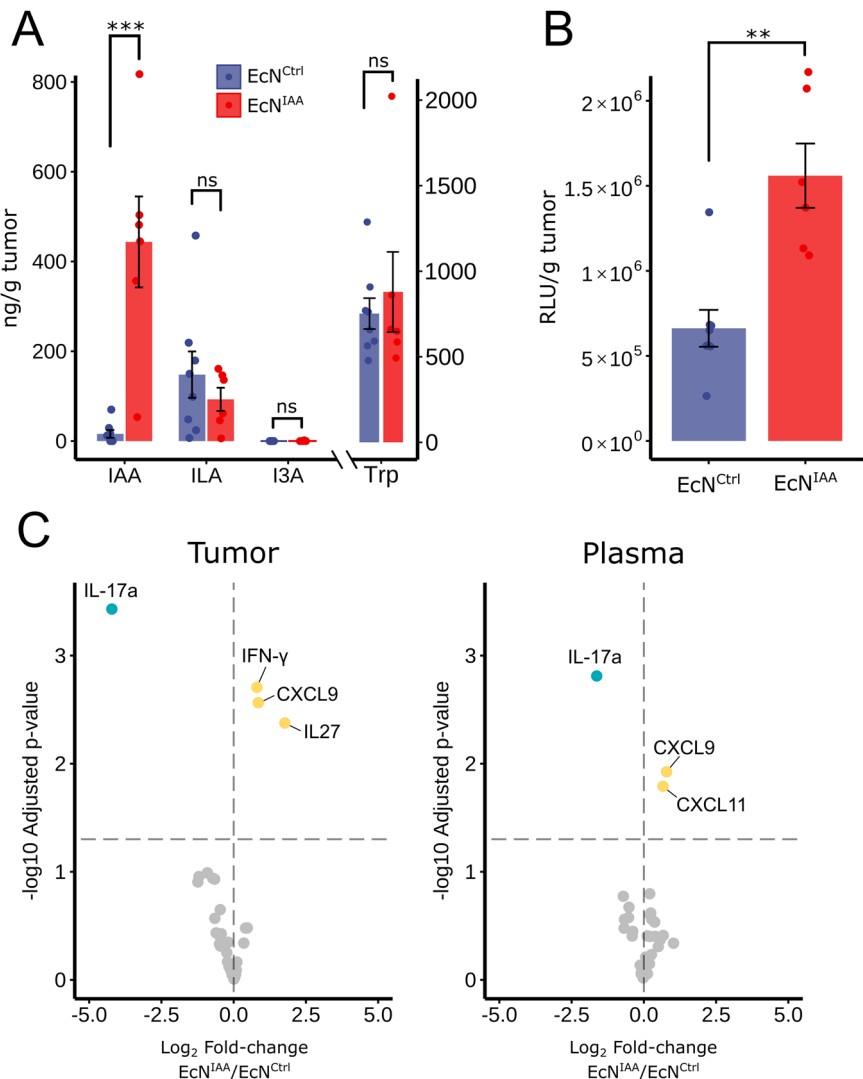

**Figure 4. Elevated levels of IAA in tumors activate AhR and are associated with changes in systemic and local cytokine expression.**

(A) Quantification of indole derivatives and tryptophan was performed on tumor homogenates using liquid chromatography–high-resolution mass spectrometry to determine their concentrations. ILA, indole-3-lactic acid ($n \geq 6$ biological replicates). (B) AhR activation was quantified by measuring the relative luminescence units (RLU) produced by luciferase-expressing AhR reporter cells stimulated for 48 h with 20% tumor homogenates from animals treated with EcN[IAA] or EcN[Ctrl] ($n \geq 6$ biological replicates). (C) Differential cytokine abundances in tumor tissue and plasma are displayed in volcano plots as log2 fold changes, comparing mice treated with EcN[IAA] vs. EcN[Ctrl]. Statistically significant downregulated cytokines are indicated in blue, and upregulated cytokines in yellow. Data are mean ± SEM. Statistical significance was determined using ANOVA, and post hoc comparisons between groups was performed using Tukey's honest significant difference test (A), two-tailed Student's $t$ test (B), and two-tailed Student's $t$ test with Bonferroni–Holm correction for multiple comparisons (C). **$P \leq 0.01$, ***$P \leq 0.001$, (A: $P = 3.59e-4$, B: $P = 0.003191$). Source data are available online for this figure.

assessed the capacity of tumor homogenates to activate AhR in an ex vivo assay. We found a significant increase in the tumor levels of IAA in the EcN[IAA]-treated group, whereas the concentrations of indole-3-lactic acid and I3A did not differ from the control group. Tumor homogenates from animals treated with EcN[IAA] exhibited significantly higher AhR activation (Fig. 4B), indicating that EcN[IAA] produces biologically relevant levels of a microbially derived AhR agonist within the tumor microenvironment.

Using a targeted proteomics assay, we quantified the levels of 48 cytokines from plasma and tumor samples. We found that animals treated with EcN[IAA] exhibited significantly elevated levels of IFN-γ, CXCL9, and IL27 within tumors, alongside reduced IL-17A

concentrations (Fig. 4C, left panel). Correspondingly, plasma analysis revealed a similar trend for CXCL9 and IL-17A, with the additional finding of increased levels of the chemokine CXCL11 (Fig. 4C, right panel). CXCL9 and CXCL11 are selective ligands for CXCR3, a receptor expressed by several types of immune cells and serve as critical mediators of immune cell migration, differentiation, and activation (Tokunaga et al, 2018). CXCL9 is crucial for recruiting activated Th1 and NK cells to focal sites and is an essential driver of T-cell-mediated suppression of cutaneous tumors (Gorbachev et al, 2007). These chemokines are primarily produced by monocytes, endothelial cells, fibroblasts, and cancer cells in response to IFN-γ (Tokunaga et al, 2018), which emphasizes the significant interaction

between immune cells, cancer cells, and stromal cells within the complex architecture of the tumor microenvironment. In CRC, IL-17A promotes tumorigenesis (De Simone et al, 2013; Song et al, 2023) and high expression is linked to poor prognosis in cancer patients. Importantly, IFN-γ[+] T cells were shown to be necessary for anti-tumor immunity in response to intratumoral delivery of I3A, the effect of which was abrogated by loss of AhR signaling (Bender et al, 2023).

These results demonstrate that EcN can serve as an effective delivery vehicle for IAA, achieving localized release within the tumor microenvironment. This delivery not only modulates cytokine levels within the tumor itself but also induces systemic effects, as shown by the altered cytokine profiles observed in plasma. These changes suggest that EcN-mediated IAA production can reshape the tumor microenvironment to favor an anti-tumor immune response while influencing broader immune signaling pathways, underscoring its potential as a therapeutic strategy for cancer immunomodulation.

## Intratumoral delivery of IAA drives anticancer immunity in the MC38 model of colorectal cancer

To evaluate whether the effects observed in the CT26 model could be generalized to other animal models of CRC, we tested the impact of EcN[IAA] in the MC38 immunocompetent model, which is syngeneic to C57Bl/6 mice (Fig. 5A). The MC38 model is characterized by a high degree of immune cell infiltration but exhibits a distinct composition and phenotype of myeloid, lymphoid, and stromal cells compared to the CT26 model (Carretta et al, 2023). This distinction is critical, as it allows evaluation of whether the therapeutic impact of EcN[IAA] is consistent across diverse tumor microenvironments with varying immune and stromal cell interactions.

Treatment with EcN[IAA] resulted in a significant decrease in tumor volume compared to the control group (Fig. 5B), as well as a significant improvement in overall survival (Fig. 5C). As with the CT26 model, the administration of bacteria was well tolerated (Fig. 5D) and EcN effectively colonized the tumor with no off-target colonization identified in the liver at euthanization (Fig. 5E). We observed no significant difference in liver and spleen weights between the groups (Fig. 5F). Almost half of the animals (5/12) treated with EcN[IAA] were tumor-free after nine days of treatment. This prompted us to test whether the treatment was associated with lasting immunity against MC38. Upon rechallenge, none of the animals previously treated with EcN[IAA] developed tumors ($n = 5$). In contrast, all ($n = 6$) treatment-naïve age-matched mice developed tumors within one week (Fig. 5G). We monitored the animals treated with EcN[IAA] for tumor development for 6 weeks post-engraftment, but no animals developed palpable tumors. This indicates treatment-associated long-term immunity, although recent studies have raised concerns about the reliability of tumor rechallenge studies as a measure of long-term immunity (Alicke et al, 2020).

Together, these results indicate that treatment with EcN[IAA] and local delivery of IAA can induce a strong and lasting immune response in multiple syngeneic tumor models.

Previous studies have provided insight into the beneficial effects of microbially derived indole metabolites in the context of cancer and established the AhR as an essential axis of communication between the microbiome, intratumoral bacteria, and the immune system (Bender et al, 2023). Our study demonstrates that this microbial axis can be harnessed using a genetically engineered chassis strain, *E. coli* Nissle 1917, which has been proven safe in clinical trials (Luke et al, 2023), to create an effective and safe microbial cancer therapeutic. While *E. coli* has previously been engineered to produce IAA in situ (Kouno et al, 2023; Wu et al, 2021), we significantly increased the production titers compared to previous studies and focused on utilizing the probiotic *E. coli* Nissle 1917 strain, which is more compatible with health-related applications. Our work focuses not only on significantly enhancing IAA production in EcN but also on applying it as an anticancer therapy, demonstrating the role of IAA in modulating tumor immunity and suppressing tumor growth.

We observed improved anti-tumor activity and survival in two syngeneic CRC mouse models after using intratumoral injection to introduce an EcN strain producing the AhR ligand IAA into tumors. We saw durable remission and clearance of bacteria after tumor remission. We investigated the mode of action using orthogonal histology and flow cytometry methods and showed that treatment with EcN[IAA] influenced the immune cell composition in the tumor microenvironment, favoring a milieu associated with higher infiltration of CD4[+] and CD8[+] cells. Higher levels of CD8[+] lymphocyte infiltration in tumors have been consistently associated with improved overall survival in multiple CRC patient cohorts (Zhao et al, 2019). We also observed a notable decrease in IL-17A following treatment with EcN[IAA], suggesting a promising avenue for combined therapy. Specifically, combining AhR modulation with immune checkpoint inhibitors could potentially amplify their efficacy, building on existing evidence that IL-17A blockade enhances immune checkpoint inhibitor responses in microsatellite stable CRC (Liu et al, 2021). Future work should explore combining other treatment modalities with EcN[IAA], such as the co-delivery of checkpoint inhibitors already established in other microbially based therapeutic platforms (Gurbatri et al, 2020).

By employing two different immune-competent models with distinct immune cell compositions, we provide proof of concept that this AMT can rewire tryptophan metabolism in solid tumors and effectively drive tumor immunity in various settings. However, it is worth noting that both models are generally considered highly immunogenic and intrinsically responsive to immune checkpoint inhibitors (Jin et al, 2022). Future work should aim to determine whether similar effects would be observed in less immunogenic ("cold") tumor models, which better reflect the subset of patients resistant to immune checkpoint inhibitors.

Immune checkpoint inhibitors have shown impressive single-agent activity in a subset of cancer patients, but they are also associated with a substantial risk of immune-related adverse events. Approximately 70–90% of patients receiving anti-PD-1/PD-L1 therapies and up to 90% of those receiving anti-CTLA-4 therapies experience adverse events (Yin et al, 2023), with combination therapies showing even higher rates of adverse events compared to monotherapies (Yin et al, 2023). While the mechanisms underlying these adverse events are not fully understood, it is generally believed that they are a side effect of disturbing immune tolerance in tissues outside of cancer (Martins et al, 2019). Our technology offers the opportunity to initiate a local immune response only within cancerous lesions. This localized approach to immunotherapy could potentially reduce toxicities associated with systemic therapy.

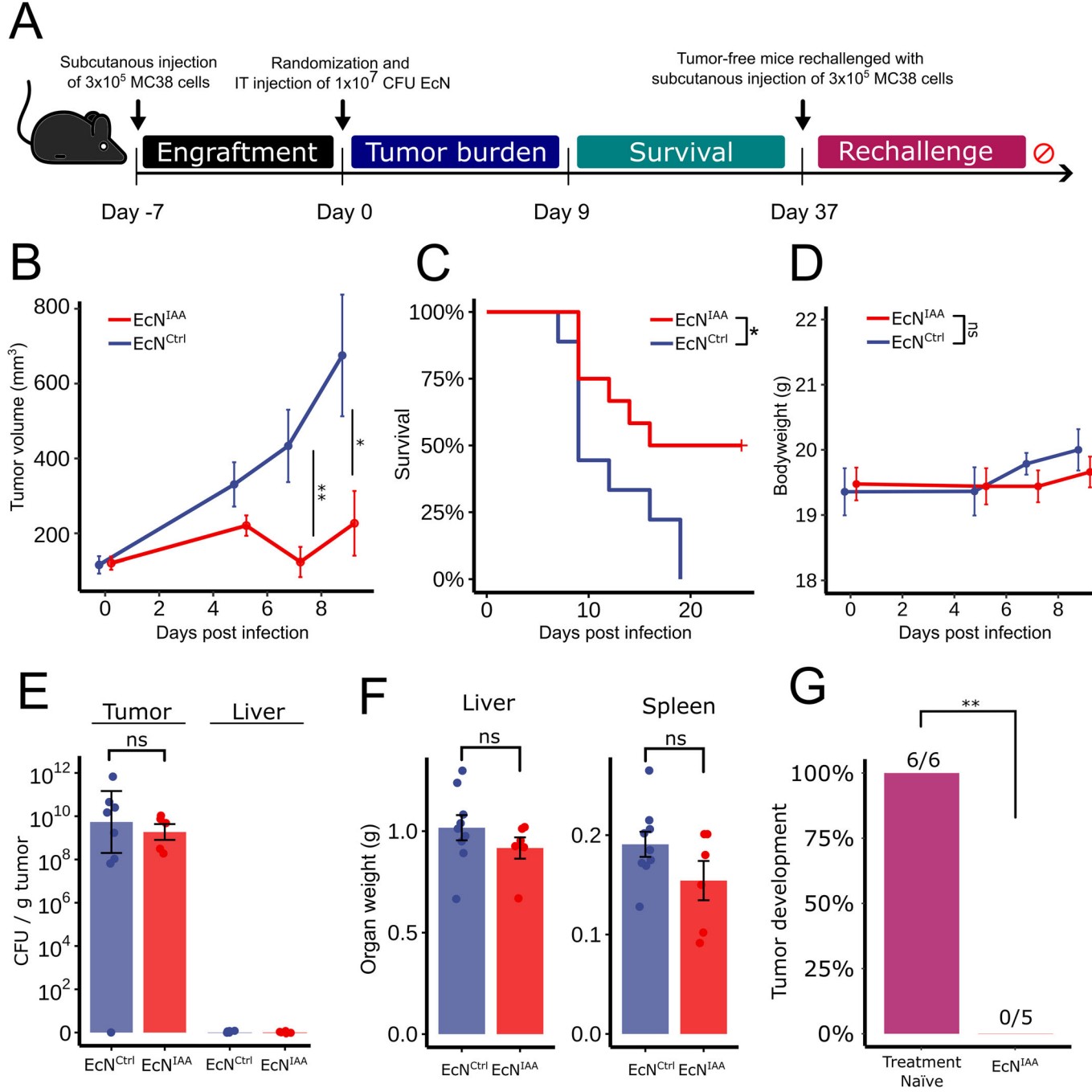

**Figure 5. Treatment with EcN^IAA shows efficacy in an additional model of CRC and is associated with lasting immunity.**

(A) Schematic representation of the experimental use of the MC38 model. Seven-week-old female C57BL/6 mice were subcutaneously implanted with MC38 cells on their right flank ($n \geq 9$ per group). Once tumors reached 75–200 mm³, mice were randomized and intratumorally injected with $10^7$ colony-forming units (CFU) EcN^Ctrl or EcN^IAA in 20 µL PBS (Day 0). Tumor size was measured three times a week using a caliper, and survival was monitored until day 37, after which all remaining animals were tumor-free. (B) Mean tumor trajectories. (C) Kaplan–Meier survival analysis censored at 25 days. (D) Bodyweight trajectories. (E) Bacterial colonization in tumors quantified as CFU per gram of tumor. Individual points represent the mean of each animal calculated from four technical replicates. (F) Liver and spleen weights. (G) Rechallenge with MC38 cells subcutaneously implanted in animals that were tumor-free following treatment with EcN^IAA and age-matched treatment-naïve C57BL/6 mice ($n \geq 5$ per group). Data are mean ± SEM. Statistical significance was determined using Wilcoxon rank-sum on day seven and the final observation timepoint preceding the first euthanasia according to humane endpoints (B), log-rank test (C), and Fisher's exact test (G), *$P \leq 0.05$, **$P \leq 0.01$. (B: day 7 $P = 0.00917$, day 9 $P = 0.03$, C: $P = 0.021$, G: $P = 0.00216$). Source data are available online for this figure.

# Methods

### Reagents and tools table

| Reagent/resource | Reference or source | Identifier or catalog number |
|---|---|---|
| **Experimental models** | | |
| CT26 murine colorectal carcinoma cell line | ATCC | CRL-2638 |
| MC38 murine colorectal carcinoma cell line | Professor Janine Erler, Copenhagen University | |
| HT29-Lucia AhR reporter cells | InvivoGen | ht2l-ahr |
| BALB/c mice | Janvier | |
| C57BL/6 mice | Taconic | |
| **Recombinant DNA** | | |
| Cas9 plasmid (pHM-156) | This study | |
| gRNA plasmid (pMB-TnaA) | This study | |
| **Antibodies** | | |
| FOXP3 | Abcam | ab215206 |
| ELA2 Neutrophil Elastase | Abcam | ab315901 |
| CD68 | Abcam | ab283654 |
| Granzyme B | Abcam | ab255598 |
| CD4 | Abcam | ab183685 |
| CD8 | Cell Signaling | D4W2Z |
| CD45-BUV395 Clone 30-F11 | BD Biosciences | #564279 |
| IFNγ-BV421 Clone XMG1.2 | BD Biosciences | #563376 |
| CD4-FITC Clone RM4-5 | BD Biosciences | #553047 |
| GzmB-PE Clone GB11 | BD Biosciences | #561142 |
| CD3-PECF594 | BD Biosciences | #562332 |
| CD8-APC Clone 53-6.7 | BD Biosciences | #553035 |
| Dead cells – eFluor 780 | Thermo Fisher Scientific | #65-0865-18 |
| **Chemicals, enzymes, and other reagents** | | |
| Q5 High-Fidelity DNA Polymerase | New England Biolabs | M0541S |
| Gibson Assembly Kit | New England Biolabs | E5510 |
| One Shot™ TOP10 Chemically Competent *E. coli* | Thermo Fisher Scientific | C4040-10 |
| NucleoSpin Plasmid EasyPure | Macherey Nagel | 740727.250 |
| Collagenase IV | Sigma-Aldrich | 17104019 |
| DNase I | Sigma-Aldrich | 11284932001 |
| TrypLE Express | Thermo Fisher Scientific | 12604013 |
| Zeocin | InvivoGen | ant-zn-05 |
| Fetal bovine serum | Thermo Fisher Scientific | A5670701 |
| Penicillin–streptomycin | Thermo Fisher Scientific | 15140148 |
| QUANTI-Luc™ Luciferase Detection Reagent | InvivoGen | rep-qlc4r2 |
| RPMI1640 | Thermo Fisher Scientific | 11875093 |
| **Software** | | Version |
| RStudio | CRAN | 4.1.0 |

| Reagent/resource | Reference or source | Identifier or catalog number |
|---|---|---|
| FlowJo® | BD Biosciences | 10.10 |
| **Other** | | |
| Synergy H1 plate reader | BioTek | |
| GentleMACS C Tubes | Miltenyi Biotec | 130-093-237 |

## Strain construction

To generate a *tnaA* knockout (EcN Δ*tnaA*), a cassette of ~1100 base pairs, inclusive of homologous regions flanking the native *tnaA* gene, was synthesized. This cassette was employed to disrupt the open-reading frame of the specified gene using seamless lambda red-mediated integration of double-stranded DNA, complemented by CRISPR/Cas9 counterselection, as delineated in prior studies (Li et al, 2015). The tnaA knockout was confirmed using primers flanking the insertion site of the cassette and Sanger sequencing. The plasmids constructs were assembled via Gibson Assembly, using *E. coli* Top 10 as the cloning host. Promoters, ribosome binding sites, and genes of interest were amplified using Q5 High-Fidelity DNA Polymerase (New England Biolabs). The assembly process followed the supplier's protocol (Gibson Assembly Protocol E5510), with 1.5 μL of the Gibson mixture used to transform One Shot™ TOP10 Chemically Competent *E. coli* (Thermo Fisher Scientific) following the supplier's guidelines (One Shot™ TOP10 Thermo Fisher Scientific catalog numbers C4040-10). Transformed cells were selected on antibiotic-containing media, and successful recombinants were verified by PCR and sequencing. A positive clone was cultured in 2 mL lysogeny broth (LB) containing 50 μg/mL kanamycin for 16 h. After incubation, plasmids were isolated using the NucleoSpin Plasmid EasyPure extraction kit (Macherey Nagel, reference number 740727.250). The purified plasmid was used for transformation of electrocompetent *E. coli* Nissle 1917 cells (Tn7:sfGFP +, StrepR, Δ*tnaA*) via electroporation. The Super-folder green fluorescent protein (sfGFP) gene was genomically integrated at the attTn7 site, spanning positions 2672041 to 2672355, and expressed under the BBa_J23101 promoter. The EcN strains used in this study were cured of any native pMUT1 plasmid. Comprehensive plasmid maps and associated sequences are included in Fig. EV1.

## Targeted high-resolution LC-HRMS

Reference solutions for the tryptophan derivatives and internal standard solutions were prepared at 1 mg/ml. These analytes were mixed and diluted with 10% ethanol in MilliQ water to prepare 0.5 μg/ml, 1 μg/ml, 5 μg/ml, 10 μg/ml, 50 μg/ml, 100 μg/ml, and 200 μg/ml solutions. All reference solutions and sample preparations included an internal standard at a fixed concentration of 4 μg/ml. Calibration curves were constructed from these mixed standards.

For analysis, 2 μL from each sample (spent medium from bacterial fermentation or tumor homogenate) was introduced into a high-efficiency liquid chromatography quadrupole time-of-flight mass spectrometry system, which included a Dionex Ultimate 3000 RS liquid chromatograph (Thermo Fisher Scientific, CA, USA)

connected to a Bruker maXis time-of-flight mass spectrometer with an electrospray interface (Bruker Daltonics, Bremen, Germany), functioning in negative ion mode. Separation of analytes was achieved on a Poroshell 120 SB-C18 column sized at 2.1 × 100 mm with 2.7 µm particle diameter (Agilent Technologies, CA, USA), employing specific parameters as recommended previously (Want et al, 2010). The column temperature was maintained at 40 °C, with the autosampler kept at 4 °C. The mobile phases for UPLC consisted of water with 0.1% formic acid (solvent A) and acetonitrile with 0.1% formic acid (solvent B). The elution of analytes started with 1% of solvent B for the initial minute, increased through a linear gradient to 15% at 3 min, reached 50% at 6 min, and peaked at 95% solvent B at 9 min. This gradient was held until the 10-min mark, after which the solvent mix was returned to starting conditions by 10.1 min, followed by a re-equilibration period until 13 min. The elution was performed at a steady flow rate of 0.4 mL/min. Mass spectrometry data acquisition was acquired in full scan mode at a frequency of 2 Hz across a scan range from 50 to 1000 $m/z$. The electrospray interface was set to the following parameters: nebulizer pressure at 2 bar, dry gas flow at 10 L/min at a temperature of 200 °C, and capillary voltage at 4500 V. For enhanced measurement precision, calibration was executed externally and internally using sodium formate clusters (Sigma-Aldrich, Schnelldorf, Germany), complemented by a lock-mass calibration (hexakis 1H,1H, 2H-perfluoroetoxy) phosphazene, Apollo Scientific, Manchester, UK).

## Cell culture and maintenance

The murine colorectal carcinoma cell line CT26 was purchased from ATCC (Catalog number: CRL-2638) and MC38 cell line was kindly provided by Professor Janine Erler, Copenhagen University. Luciferase-labeled HT29-Lucia cells, expressing luciferase under the control of the Cyp1a1 promoter (referred to as AhR reporter cells), were obtained from InvivoGen (catalog reference ht2l-ahr). CT26 and MC38 cell lines were grown in RPMI1640 medium (Thermo Fisher Scientific, catalog number 11875093) and HT29-Lucia cells in McCoy's 5A (Modified) medium (Thermo Fisher Scientific, catalog number 16600082). Media were supplemented with 10% fetal bovine serum (Thermo Fisher Scientific, catalog number A5670701) and 1% penicillin–streptomycin (Thermo Fisher Scientific, catalog number 15140148) in a humidified incubator at 37 °C with 5% $CO_2$. Cells were maintained by changing the media twice per week and passaged upon reaching 90% confluence to ensure optimal growth conditions and cellular health. All cell lines were tested for mycoplasma contamination to ensure their integrity and suitability for experimental use.

## Preparation of bacterial supernatants

Glycerol stocks of EcN were inoculated into LB with 100 µg/mL streptomycin and 50 µg/mL kanamycin and grown overnight at 37 °C with shaking at 250 RPM. Overnight cultures were diluted 1:100 in fresh medium and grown for 24 h. Cultures were centrifuged at 4000 × $g$ for 10 min, and supernatants were filter-sterilized using a 0.22 µm syringe filter. Supernatants were stored at −20 °C untill use. Bacterial supernatants for the AhR activity assay were prepared fresh and used immediately to avoid potential freeze–thaw effects.

## AhR activity assay

HT29-Lucia™ AhR Cells (InvivoGen catalog number ht2l-ahr) were handled according to the manufacturer's instructions. Briefly, cells were thawed and passaged for two generations without antibiotic selection with zeocin (InvivoGen CAS number 11006-33-0). Subsequent passages were maintained in 100 µg/mL zeocin and 1% penicillin–streptomycin. Cells were harvested before reaching 90% confluence, and ~5000 cells per well were seeded in 96-well plates (Greiner bio-one catalog number 655160) in 160–180 µL volume. Samples or positive controls (20–40 µL) was then added to a final volume of 200 µL and incubated for 48 h. Following incubation, 20 µL of stimulated cell supernatant was transferred to a black 96-well plate with an optical bottom (Thermo Fisher Scientific catalog number 165305). To each well, 50 µL QUANTI-Luc™: Luciferase Detection Reagent (InvivoGen catalog number rep-qlc4r2) was added, and luminescence was immediately read on a Synergy H1 plate reader (BioTek, Winooski, VT, USA) using 2 mm read-height, 200 ms integration time, and 100 gain.

## Animal experiments

The animal experiments were conducted in accordance with the Danish Animal Experiments Act for the protection of animals used for scientific purpose (LBK 1107 from 02/07/2022) and Directive 2010/63/EU of the European Parliament. Moreover, the study protocols were approved by the Animal Experimentation Committee under the Ministry of Food, Fishing, and Agriculture (license number 2021-15-0201-00925). Thirty female BALB/c mice (Janvier) 7 weeks old and 35 female C57BL/6 (Taconic) 7 weeks old underwent a minimum of 5 days of acclimatization with ad libitum access to tap water and chow diet (A30, Safe diets). At the start of the study, $3 \times 10^5$ CT26 (BALB/c mice) and $3 \times 10^5$ MC38 (C57BL/6 mice) tumor cells in 100 µl serum-free media were subcutaneously engrafted on the right hind flank. Tumors were grown to an average volume of 50–200 mm³ before the mice were randomized into two groups (n ≥ 9). Animals received $1 \times 10^7$ CFU bacteria in 20 µl PBS by intratumoral injection. Tumors were assessed three times per week using caliper, and the volume was estimated using the formula 0.5× (*Longest dimension × shortest dimension²*). A rechallenge with $3 \times 10^5$ MC38 tumor cells in 100 µl serum-free media was performed for five remaining tumor-free animals following treatment with EcN[IAA] and six age-matched treatment-naïve C57BL/6 mice as controls. As the studies progressed mice would be humanely euthanized using cervical dislocation when tumor volumes reached 2000 mm³ or fluid ulceration appeared on the tumor. Whole blood was collected upon euthanizing and immediately processed using BD Biosciences Microtainer™ Tubes with Microgard™ Closure (Thermo Fisher Scientific product code 12957646) by centrifugation at 9000×$g$ for 2 min. Plasma was collected and immediately stored at −20 °C for further use. Moreover, organ weights for liver and spleen, as well as tumor weights, were recorded. In addition, tumors and livers were processed according to description below for generation of homogenates. The mice were co-housed in groups of 4–6 in Individually Ventilated Cages (IVC), 22 °C ± 2 °C, with a 12-hour light/darkcycle (06:00 to 18:00). The mice were given ad libitum access to water and chow diet (A30, Safe diets) during the whole study period. The animal studies were carried out as single-blinded trials, with measurements and ulceration scoring conducted by an animal caretaker who was blinded to group allocation.

## Preparation of tumor and liver homogenates and determination of colony-forming units

After dissection, uniformly sized tumor sections or whole livers were placed into GentleMACS C tubes and homogenized using the *m_Imptumor_01_01* program. The homogenized samples were then filtered through a 70-µm cell strainer to remove cell clumps. Serial dilutions of the tumor and liver homogenates were plated on LB agar plates containing 100 µg/mL streptomycin and 50 µg/mL kanamycin, and bacterial colonies were counted the following day to estimate the colony-forming units (CFU).

## Preparation of plasma from whole blood

Whole blood was collected upon euthanasia and immediately processed using BD Biosciences Microtainer™ Tubes with Microgard™ Closure (Thermo Fisher Scientific product code 12957646) by centrifugation at 9000×g for 2 min. Plasma was collected and immediately stored at −20 °C for further use.

## Cryopreservation of tumors

Half the tumor from each animal was cryopreserved according to (Liang et al, 2021) Briefly, tumors were cleaned and sectioned into 3 × 3 mm pieces using a scalpel. These pieces were then submerged in 1 mL of storage medium consisting of 90% fetal bovine serum (Thermo Fisher Scientific catalog number A5670701) and 10% dimethyl sulfoxide for snap-freezing and cryopreservation in liquid nitrogen.

## Flow cytometric analysis of tumor-infiltrating lymphocytes

The cryopreserved portion of tumors was thawed in a 37 °C water bath, and tissues were immediately transferred to a 15-mL tube and washed in 10 mL RPMI1640 medium with 10% fetal bovine serum. After thawing, biopsies were further cut into smaller pieces on a TES-99 cooling surface at 4 °C and incubated in a digestion medium of RPMI1640 supplemented with 200 U/mL collagenase IV (Sigma-Aldrich catalog number 17104019) and 10 U/mL DNAse I (Sigma-Aldrich catalog number 11284932001) for 30 min at 37 °C with shaking. The digestion suspension was centrifuged at $400 \times g$, 4 °C, and the pellet was resuspended in TrypLE Express (Thermo Fisher Scientific catalog number 12604013) to yield a single-cell suspension.

All subsequent steps were performed on ice when possible. Single-cell suspensions were filtered with a 70-µm cell strainer into a 50-mL collection tube, and the filter was washed repeatedly using PBS. Suspensions were centrifuged at 400×g 4 °C, and pellets were resuspended in FACS buffer. Cells were Fc blocked with anti-mouse CD16/CD32 (BD Biosciences) for 5 min on ice and then stained for extracellular markers (CD45, CD3, CD4, CD8). For intracellular staining (IFN-γ, GzmB), cells were fixed using a 1% paraformaldehyde solution in PBS and then stained in a 0.1% (w/v) saponin permeabilization buffer. Data acquisition was performed on an LSRFortessa™ X-20 (BD Biosciences) using the FACSDiva™ Software 9.0 (2019) (BD Biosciences), and compensation was carried out using a single-color-stained AbC™ Total Antibody Compensation Bead Kit (Invitrogen catalog number A10497) and ArC Amine Reactive beads (Invitrogen catalog number A10346). Analysis of the flow cytometry data was conducted using FlowJo® version 10.10 (BD Biosciences). Fluorescence minus one control was employed to establish gating for the markers (Fig. EV4). Only viable cells were included in the analysis, as determined by gating based on staining with the viability dye eFluor™ 780 (Thermo Fisher Scientific catalog number 65-0865-18). Full list of antibodies used for extra- and intracellular targets in the panel are listed in the Reagents and Tools Table.

## Immunohistochemical and bacterial analysis of tumors

Tumor biopsies were immediately fixed in 4% buffered paraformaldehyde (pH 7.4) and stored at 4 °C for a minimum of 24 h prior to embedding in paraffin. Formalin-fixed paraffin-embedded blocks were sectioned into 3–5 µm thick sections for subsequent immunohistochemical (Granzyme B, CD4, and CD8) and bacterial (Universal bacterial probe) analysis. Sections intended for immunohistochemical analysis were microwaved for 15 min in Tris-EGTA buffer at pH 9 for antigen retrieval. This was followed by a pre-incubation in 2% bovine serum albumin for 10 min and an incubation at room temperature for one hour with primary antibodies diluted in 2% bovine serum albumin: FOXP3 (Abcam catalog number ab215206, made in rabbit, diluted 1:250, ELA2 Neutrophil Elastase (Abcam catalog number ab315901), made in rabbit, diluted 1:2000, CD68 (Abcam catalog number ab283654), diluted 1:100, Granzyme B (Abcam catalog number ab255598), made in rabbit, diluted 1:3200; CD4 (Abcam catalog number ab183685), made in rabbit, diluted 1:500; and CD8 (Cell Signaling catalog number D4W2Z), made in rabbit, diluted 1:250. Sections were incubated for 40 min with biotinylated secondary antibody immunoglobulins goat anti-rabbit BA-1000 diluted 1:200 to amplify the reaction. Subsequently, 3% hydrogen peroxide was used to quench endogenous peroxidase. A third incubation step was conducted with a preformed avidin and biotinylated horseradish peroxidase macromolecular complex (for CD4 and granzyme B: Vector Laboratories Elite ABC code number PK-6100; for CD8a, ABC code number PK-4000) for 30 min. The reaction was visualized using 3,3-diaminobenzidine (DAB+) (for granzyme B, Vector Laboratories code number SK-4105; for CD4 and CD8a, SK-4100) for 15 min, and sections were counterstained with Mayer's hematoxylin.

Fluorescence in situ hybridization (FISH) was used to detect the presence and distribution of *E. coli* within tumor sections using a universal (BacUni) bacterial probe (AdvanDx, Woburn, MA) tagged at the 5' end with Texas Red. FISH staining was carried out as previously described (Kvich et al, 2024). Bacteria were visualized with a high-speed automated imaging system (GeoMx® Digital Spatial Profiler, NanoString).

## Plasma and tumor biomarker measurements

Plasma and tumor biomarkers were determined using proximity extension assay technology (Olink Proteomics, Inc.) on an Olink Target 48 Mouse Cytokine Panel. Biomarkers with more than 20% of measurements outside the assay limit of detection were excluded from the analysis.

## Statistical testing

Statistical analyses were conducted using *RStudio version 4.1.0*, using the *rstatix* and *DescTools* packages. Data are presented as mean ± SEM unless otherwise specified. *P* values of <0.05 were considered statistically significant. A dependent sample *t*-test or paired Wilcoxon-signed rank test was employed for comparisons between the two groups. In cases of multiple comparisons, Tukey's honestly significant difference test or Benjamini–Hochberg procedure adjustments were performed.

## Data availability

No primary datasets have been generated and deposited.

The source data of this paper are collected in the following database record: biostudies:S-SCDT-10_1038-S44319-025-00386-9.

## Peer review information

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

## Acknowledgements

This work received funding from The Novo Nordisk Foundation under NNF grant number: NNF20CC0035580, the NNF Challenge program CAMiT under grant agreement: NNF17CO0028232 and the NNF Pioneer Innovator 2-2021, Immunomodulation of the tumor microenvironment using tumor-targeting cytokine-secreting bacteria under grant agreement NNF21OC0072832. The authors are grateful to DTU Bio Facility's animal caretakers, Heidi Arps and Anders Johannes Moustgaard, for assistance with the in vivo experiments. The authors would like to thank Heidi Marie Paulsen, Bente Stærgaard and Steen Seier Poulsen from Copenhagen University Histolab and Randi Foged Melohn from the Center for Surgical Science at Zealand University Hospital for their services and support in histological analysis. The authors are grateful to Mikael Pedersen from DTU National Food Institute for assistance in conducting the metabolomic analysis. The authors would also like to express their gratitude to Anna Viktoria Vaaben for her valuable feedback and discussion of the manuscript.

## Author contributions

**Troels Holger Vaaben**: Conceptualization; Data curation; Formal analysis; Investigation; Visualization; Methodology; Writing—original draft; Project administration; Writing—review and editing. **Ditte Olsen Lützhøft**: Methodology; Writing—original draft; Project administration; Writing—review and editing. **Andreas Koulouktsis**: Data curation; Investigation; Methodology. **Ida Melisa Dawoodi**: Data curation; Methodology. **Camilla Stavnsbjerg**: Data curation; Methodology. **Lasse Kvich**: Data curation; Methodology. **Ismail Gögenur**: Data curation; Investigation; Project administration. **Ruben Vazquez-Uribe**: Funding acquisition; Writing—original draft; Project administration; Writing—review and editing. **Morten Otto Alexander Sommer**: Conceptualization; Funding acquisition; Writing—original draft; Project administration; Writing—review and editing.

Source data underlying figure panels in this paper may have individual authorship assigned. Where available, figure panel/source data authorship is listed in the following database record: biostudies:S-SCDT-10_1038-S44319-025-00386-9.

## Disclosure and competing interests statement

THV, DLO, AK, RVU and MOAS are inventors on a patent filed by DTU. The remaining authors declare no competing interests.

# Expanded View Figures

**Figure EV1.  Schematic representation of the pMUT1 plasmid used to produce IAA.**

The plasmid contains a ColE2-like origin of replication and a kanamycin resistance cassette for selection and the Hok/Sok toxin-antitoxin system for improved stability. The genes encoding the three-step pathway were codon-optimized for *E. coli* and inserted into the plasmid as an operon, under the control of the constitutive promoter pMS6.

▶

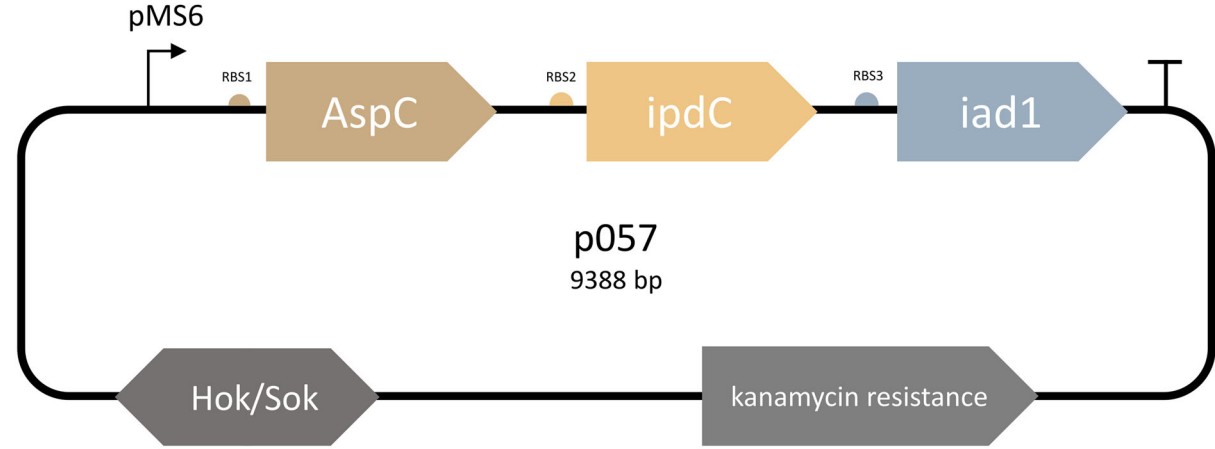

| Name | Part | Sequence (5'--> 3') |
|---|---|---|
| MS6 | Promoter | TGCTGGACTCGTCGTAATCCTGCGTGTATAATTGGC |
| RBS1 | RBS4 | AAAGGAGAA |
| AspC | Gene of interest 1 | ATGTTCGAGAATATTACTGCCGCCCCGGCTGACCCAATCTTAGGTCTTGCCGATTTGTTTCGCGCTGACGAACGGCCAGGCAAGATAAATTTGGGCATAGGCGTTTACAAGGATGAAACTGGCAAGACGCCTGTCCTGACATCCGTAAAAAAGGCTGAACAATACCTTCTTGAAAACGAGACAACAAAAGAACTATCTGGGGATTGATGGAATACCCGAATTTGGTCGGTGCACTCAAGAACTGCTTTTCGGAAAGGGGTCTGCTCTTATCAACGATAAGCGCGCCCGTACAGCACAGACACCTGGCGGAACAGGTGCGTTGAGAGTCGCAGCGGATTTTCTGGCGAAGAATACGAGCGTTAAAAGAGTGTGGGTCTCAAATCCATCGTGGCCGAACCATAAGAGCGTCTTCAACAGTGCAGGTTTAGAAGTCAGAGAGTATGCGTACTACGACGCGGAAAATCATACACTGGACTTCGACGCGCTGATAAACTCGTTGAACGAGGCACAAGCTGGAGATGTTGTACTTTTCCACGGGTGCTGTCACAACCCTACAGGAATAGATCCTACGCTTGAACAATGGCAGACCTTAGCGCAATTGTCTGTTGAAAAAGGATGGTACCGTTATTTGACTTTGCTTACCAGGGTTTTGCGCGCGGTCTGGAAGAGGATGCAGAGGGGCTGCGGGCTTTTGCAGCTATGCATAAAGAACTTATAGTGGCCAGTAGCTACTCTAAAAATTTCGGGTTATACAACGAACGGGTCGGTGCGTGTACGTTGGTTGCAGCCGACTCTGAAACAGTAGATCGTGCTTTCTCCCAAATGAAGGCAGCGATTAGAGCGAACTATTCTAATCCTCCCGCCCATGGCGCCTCGGTAGTAGCCACCATTCTTAGTAACGACGCCTTACGGGCCATATGGGAACAAGAATTGACCGATATGCGCCAGCGGATTCAGAGAATGAGACAGCTGTTCGTTAACACGCTTCAAGAAAAGGGTGCCAACCGTGATTCTCCTTTATTATCAAGCAGAATGGCATGTTCTCATTTAGTGGGCTGACGAAAGAGCAAGTGCTGCGCCTGCGTGAGGAATTCGGTGTTTACGCAGTGGCTTCGGGACGTGTCAACGTGGCAGGAATGACACCGGACAACATGGCCCCACTGTGTGAGGCAATTGTCGCCGTTCTGTAA |
| RBS2 | RBS4 | AAAGGAGAA |
| ipdC | Gene of interest 2 | ATGAGAACACCGTATTGCGTTGCAGATTACCTTTTAGATAGATTGACCGACTGTGGTGCAGATCACTTATTCGGGGTGCCTGGAGATTACAATTTACAGTTCTTAGACCATGTAATAGACTCCCCCGACATCTGTTGGGTCGGCTGTGCAAATGAATTGAACGCGAGCTACGCTGCAGACGGCTATGCGCGGTGTAAAGGGTTCGCCGCTCTTCTTACTACATTTGGCGTGGGTGAACTTTCCGCAATGAATGGTATTGCGGGGTCTTATGCCGAGCACGTTCCAGTGCTGCATATAGTAGGCGCTCCGGGAACAGCGGCTCAACAGCGTGGGGAGTTACTTCATCATACCCTGGGCGATGGAGAATTTAGACACTTTTACCACATGAGCGAACCGATTACAGTTGCTCAAGCGGTTTAACCGAGCAAAATGCCTGCTATGAAATTGATCGTGTACTTACAACCATGCTTCGTGAACGTAGACCTGGGTATTTGATGTTACCGGCCGATGTGGCGAAAAAGGCAGCTACTCCGCCTGTCAATGCCCTTACACACAAACAAGCGCATGCTGATTCGGCGTGTCTTAAGGCTTTTCGCGACGCGGCTGAAAACAAATTAGCTATGTCTAAACGGACAGCGTTATTGGCCGACTTCTTAGTGTTGCGGCACGGCCTTAAACACGCATTACAGAAGTGGGTAAAGGAAGTTCCTATGGCACACGCCACCATGTTGATGGGTAAAGGCATTTTTGATGAAAGACAGGCTGGGTTTTACGGTACTTACTCGGGTTCGGCGTCAGCACTGGTGCAGTCAAAGAAGCCATAGAGGGGGCAGACACGGTCCTGTGCGTAGGTACCAGATTTACTGACACCCTGACTGCCGGCTTTACTCACCAATTAACACCTGCACAGACTATTGAGGTCCAGCCGCACGCTGCCAGAGTGGGAGACGTTTGGTTCACTGGAATACCTATGAACCAGGCTATTGAGACACTGGTGGAACTTTGTAAGCAACATGTTCATGCAGGTCTTATGTCATCCTCGTCGGGGGCAATCCCCTTTCCCCAACCAGACGGTAGCCTTACCCAAGAGAACTTCTGGAGAACCCTGCAAACCTTCATACGCCCTGGCGATATCATCCTTGCGGACCAGGGAACGTCAGCCTTCGGGGCTATAGACCTGCGGCTGCCTGCGGACGTTAATTTCATAGTACAGCCCCTTTGGGGATCAATTGGCTATACATTAGCTGCCGCGTTTGGGGCCCAAACAGCCTGTCCAAACCGGCGTGTTATTGTCCTGACAGGCGACGGGGCTGCTCAACTGACAATCCAGGAGCTTGGAAGCATGCTTAGAGCACAAGCAACACCCCGATCATCCTTGTGTTAAACAACGAGGGCTACACCGTAGAACGGGCTATCCACGGCGCTGAGCAGCGCTACAACAGATATCGCTTTATGGAATTGGACCCACATTCCGCAAGCCCTTTCACTGGACCCGCAATCGGAATGCTGGCGTGTCTCAGAAGCCGAACAATTGGCCGACGTACTGGAGAAGGTTGCCCATCACGAGCGTTTGAGTTTAATAGAAGTTATGCTGCCGAAGGCGGATATACCACCCTTGTTGGGAGCGCTGACGAAAGCGTTGGAGGCCTGCAATAACGCGTGA |
| RBS3 | RBS3 | ACAGGAGGG |
| iad1 | Gene of interest 3 | ATGCCTACACTGAACTTGGATTTGCCGAACGGCATAAAGAGTACGATCCAGGCAGACCTTTTTATAAATAATAAGTTTGTCCCCGCGTTGGATGGGAAGACGTTTGCAACGATTAACCCTAGTACCGGCAAGGAGATAGGCCAAGTCGCGGAGGCCTCGGCGAAAGACGTTGACTTAGCGGTGAAGGCCGCCCGTGAAGCTTTTGAGACTACTTGGGGCGAAAATACGCCGGGAGACGCGCGGGGACGCTGTTGATTAAATTAGCTGAATTGGTTGAGGCAAATATAGACGAATTAGCCGCAATCGGAATGGCTGGATAACGGAAAAGCATTCTCCATTGCTAAGTCCTTTGACGTGGCAGCCGTGGCCGCAAACTTAAGATATTATGGCGGTTGGGCAGATAAGAATCATGGCAAGGTGATGGAAGTTGACACGAAAAGATTGAATTATACTCGCCACGAACCGATAGGTGTTTGTGGGCAAATCATTCCGTGGAACTTCCCACTGCTTATGTTCGCTTGGAAATTGGGCCCAGCCTTAGCCACGGGTAATACTATAGTTTTGAAAACGGCAGAACAGACGCCCTTATCAGCAATCAAAATGTGTGAGCTTATAGTAGAAGCAGGTTTTCCTCCTGGGGTAGTGAACGTCATATCTGGCTTTGGACCTGTGGCGGGCGCGGCAATTAGTCAGCATATGGACATCGATAAAATTGCCTTCACAGGTTCCACCTTAGTCGGTCGTAACATTATGAAAGCGGCAGCTAGCACGAATCTGAAAAAAGTGACCCTTGAGTTGGGTGGGAAAAAGCCCTAATATAATCTTCAAAGATGCAGACTTGGATCAAGCCGTTAGATGGAGTGCATTTGGCATTATGTTTAACCACGGGCAGTGTTGTTGTGCGGTCCCGCGTCTACGTCGAAGAAAGTATTTACGATGCCTTTATGGAGAAGATGACTGCCCATTGCAAGGCTTTACAGGTAGGAGATCCTTTCTCGGCCAATACGTTCCAGGGCCCTCAGGTGTCGCAGTTACAGTACGACCGGATTATGGAATATATAGAATCTGGCAAGAAAGATGCCAAACTCTTGATAGGTGGGGTCAGAAAAGGGAACGAGGGCTATTTTATTGAACAACTATCTTTACCGATGTTCCGCATGATGCCAAAATCGCAAAGGAAGAGATCTTCGGACCGGTTGTCGTTGTATCAAAATTTAAGGACGAAAAAGAATTGATTAGAATAGCTAACGATAGCATTTACGGATTAGCTGCGGCTGTGTTTAGTCGGGATATCTCACGCGCTATTGAGACGGCACATAAGCTTAAGGCCGGTACTGTATGGGTTAACTGTTATAACCAACTTATACCCCAAGTCCCATTCGGTGGCTACAAGGCCAGTGGGATTGGACGCGAGCTGGGTGAATAGTCTCTGTCGAACTACACGAATATCAAGGCTGTACATGTAAACTTAAGTCAGCCTGCCCCAATTTAA |
| Terminator | Terminator | AAAGGGGCTACCGGCGAACCAGCAGCCCCTTTATAAAGGCGCTTCAGTAGTCAGACCAGCATCAGTCCTGAAAAGGCGGGCCTGCGCCCGCCTCCAGG |
| KanR | Kanamycin selection marker | ATGATTGAACAGGATGGCCTGCATGCGGGTAGCCCGGCAGCGTGGGTGGAACGTCTGTTTGGCTATGATTGGGCGCAGCAGACCATTGGCTGCTCTGATGCGGCCGGTGTTTCGTCTGAGCGCGCAGGGTCGTCCGGTGCTGTTTGTGAAAACCGATCTGAGCGGTGCGCTGAACGAGCTGCAGGATGAAGCGGCGCGTCTGAGCTGGCTGGCCACCACCGGTGTTCCGTGTGCCGGCGGTGCTGGATGTGGTGACCGAAGCGGGCCGTGGATTGGCTGCTGCTGGCTGGCCGCCGGCAGAAAAGTGAGCATTATGGCGGATGCCATGCGTCGTCTGCATACCCTGGACCCGGCGACCTGTCCGTTTGATCATCAGGCGAAACATCGTATTGAACGTGCGCGTACCCGTATGGAAGCGGGCCTGGTGGATCAGGATGATCTGGATGAAGAACATCAGGGCCTGGCACCGGCAGAGCTGTTTGCGCGTCTGAAAGCGAGCATGCCGGATGCGGAAGATCTGGTGGTGACCCATGGTGATGCGTGCCTGCCGAACATTATGGTGGAAAATGGCCGTTTTAGCGGCTTTATTGATTGCGGCGTCTGGGCGTGGCGGATCGTTATCAGGATATTGCGCTGGCCACCCGTGATATTGCGGAAGAACTGGGCGGCGAATGGGCGGATCGTTTTCTGGTGCTGTATGGCATTGCGGCACCGGATAGCCAGCGTATTGCGTTTTATCGTCTGCTGGATGAATTTTTCTAA |
| Hok/Sok locus | Plasmid stability | ACAACATCAGCAAGGAGAAAGGGGCTACCGGCGAACCAGCAGCCCCTTTATAAAGGCGCTTCAGTAGTCAGACCAGCATCAGTCCTGAAAAGGCGGGCCTGCGCCCGCCTCCAGGTTGCTACTTACCGGATTCGTAAGCCATGAAAGCCGCCACCTCCCTGTGTCCGTCTCTGTAACGAATCTCGCACAGCGATTTTCGTGTCAGATAAGTGAATATCAACAGTGTGAGACACACGATCAACACACACCAGACAAGGGAACTTCGTGGTAGTTTCATGGCCTTCTTCTTCCTTGCGCAAAGCGCGGTAAGAGGCTATCCTGATGTGGACTAGACATAGGGATGCCTCGTGGTGGTTAATGAAAATTAACTTACTACGGGGCTATCTTCTTTCTGCCACACAACACGGCAACAAACCACCTTCACGTCATGAGGCAGAAAGCCTCAAGCGCGGGCACATCATAGCCCATATACCTGCACGCTGACCACACTCACTTTCCCTGAAAATAATCCGCTCATTCAGACCGTTCACGGGAAATCCGTGTGATTGTTGCCGCATCACGCTGCCTCCCGGAGTTTGTT |

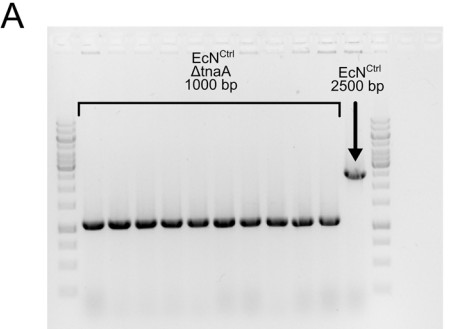

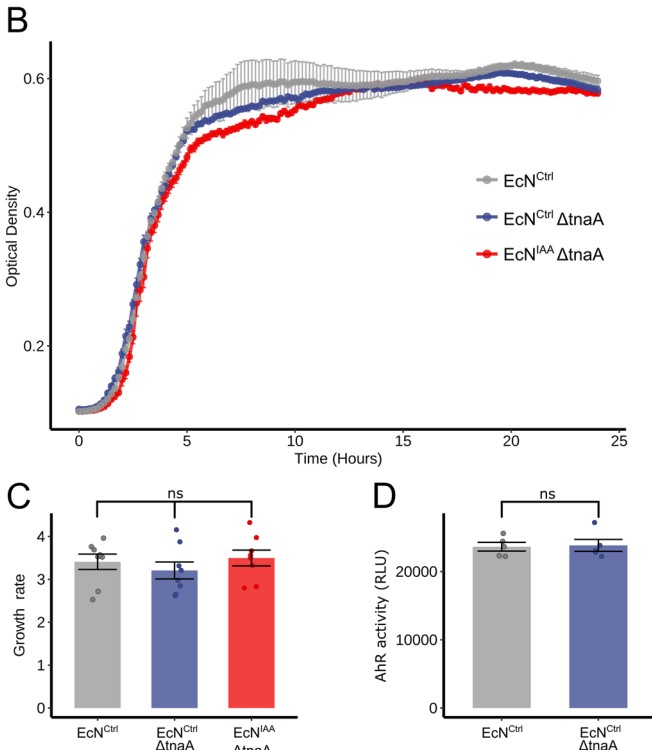

**Figure EV2. Confirmation of ΔtnaA knockout and functional characterization of *Escherichia coli* Nissle 1917 strains.**

(A) Agarose gel electrophoresis confirming the tnaA gene knockout in EcN. The wild-type control strain (EcN^Ctrl) shows a band of approximately 2500 bp, whereas the ΔtnaA knockout strain show a smaller band around 1000 bp, confirming successful deletion of the tnaA gene. (B) Growth curves of EcN strains over 24 h, measured by optical density at 600 nm ($n = 8$ biological replicates). Deletion of the tnaA gene does not significantly affect the growth rates of the strains. (C) Specific growth rate calculated from growth curves in (B). (D) AhR activity as relative luminescent units (RLU) derived from luciferase-expressing AhR reporter cells following stimulation with 10% supernatant of EcN^Ctrl and EcN^Ctrl ΔtnaA for 48 h ($n = 5$ biological replicates). Data are mean ± SEM. Statistical significance was determined with ANOVA and post hoc comparison analysis between groups using Tukey's honest significant difference test. Source data are available online for this figure.

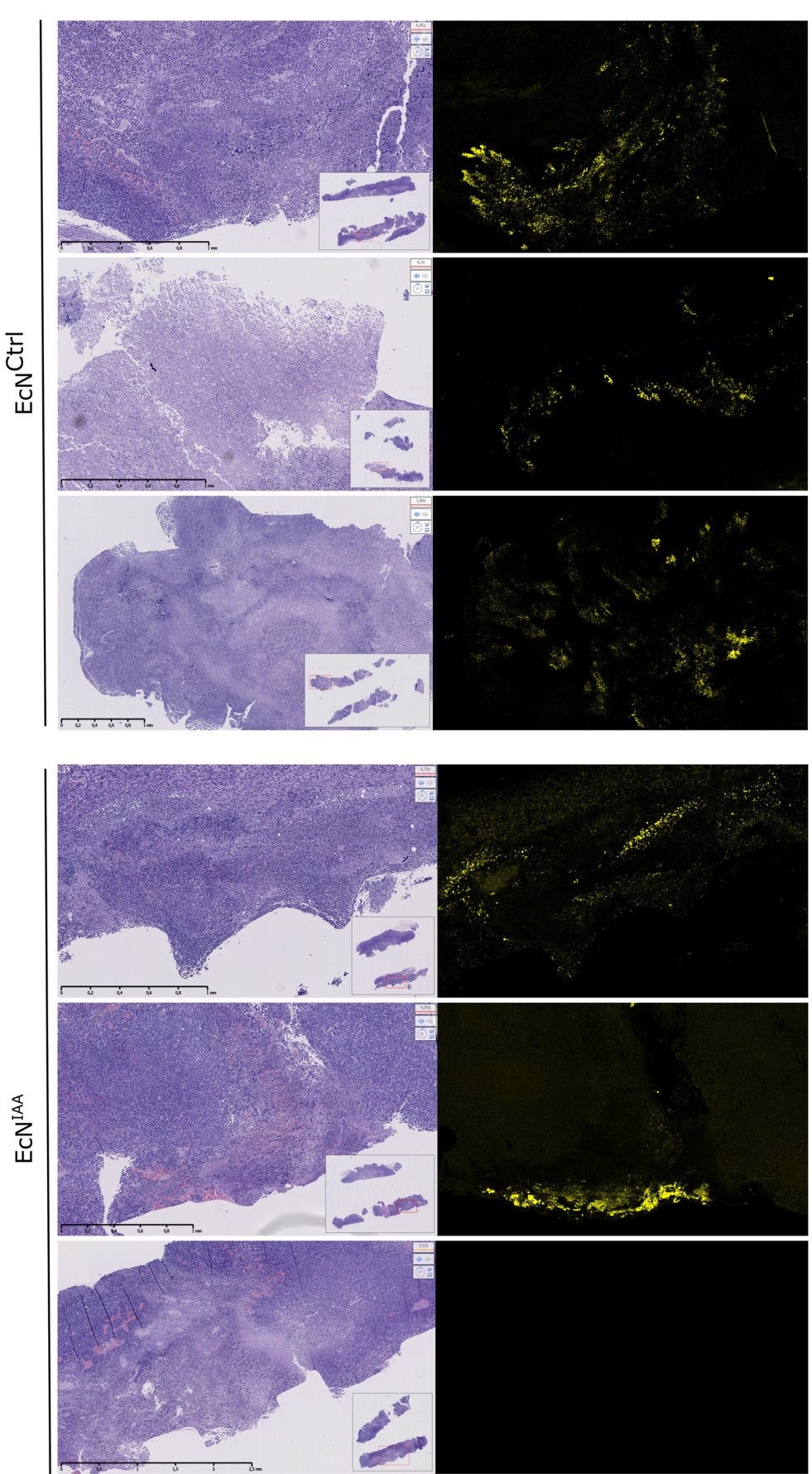

**Figure EV3. Localization of EcN in necrotic tumor regions.**

Representative images of areas with necrosis and co-localized bacteria from animal study with CT26 tumors receiving an intratumoral injection of EcN$^{Ctrl}$ (top) and EcN$^{IAA}$ (bottom). Tumors of comparable size from EcN$^{IAA}$ ($n = 3$ biological replicates) and EcN$^{Ctrl}$ ($n = 3$ biological replicates) animals were analyzed in formalin-fixed paraffin-embedded sections using H&E staining (left column) and fluorescence in situ hybridization microscopy to detect bacteria on the adjacent slide (right column).

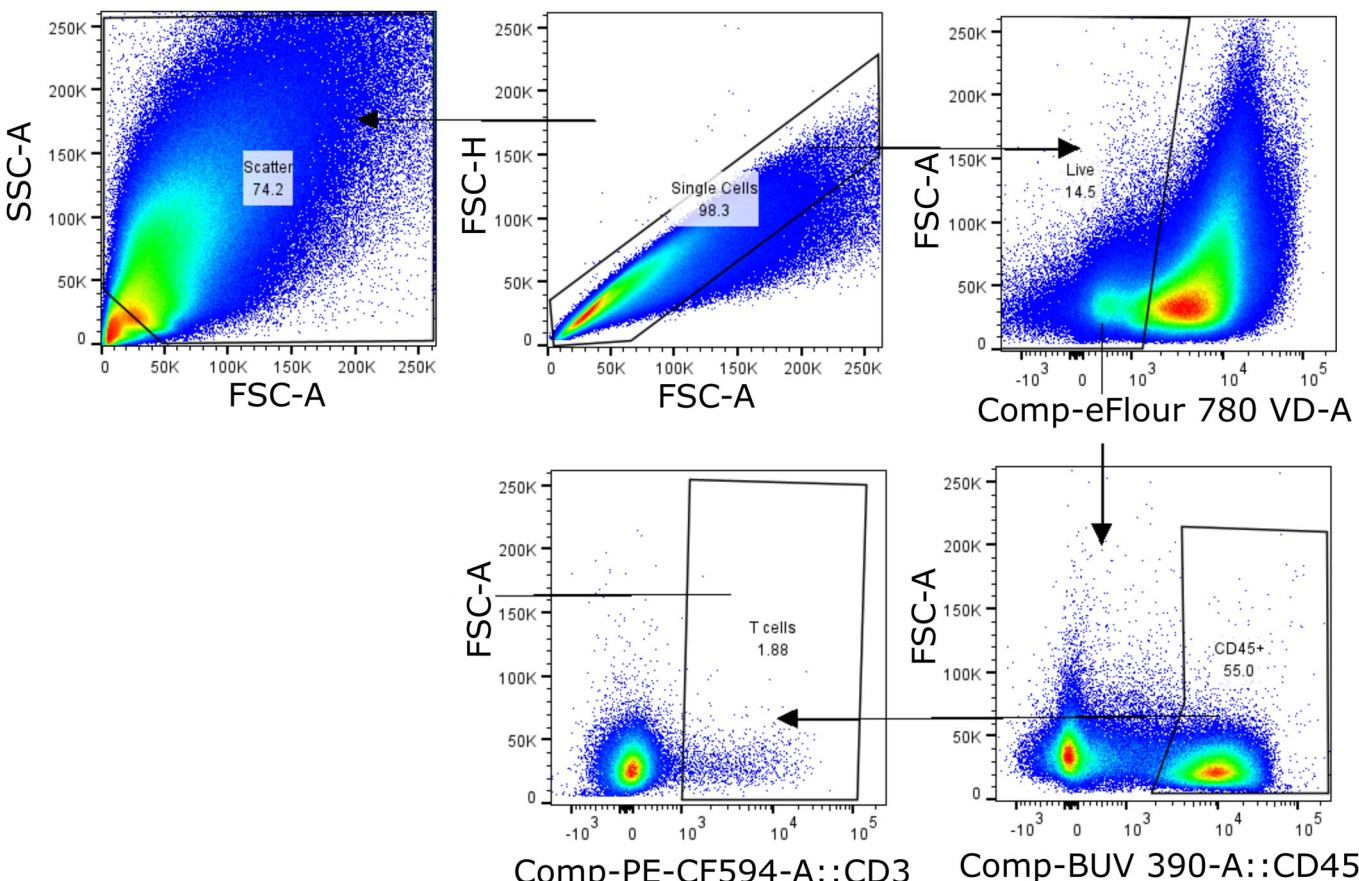

**Figure EV4.  Gating strategy for T-cell panel related to Fig. 2g.**

Identification of viable T cells was defined as viability dye (VD), CD45 +, CD3 +. Fluorescence Minus One control was used to determine gates.

