## [Peer Review File · EMBO Reports]

Modulating tumor immunity using advanced microbiome therapeutics producing an indole metabolite

Troels Holger Vaaben, Ditte Lützhøft, Andreas Koulouktsis, Ida Dawoodi, Camilla Stavnsbjerg, Lasse Kvich, Ismail Gögenur, Ruben Vazquez-Urbe, and Morten Sommer

Corresponding author(s): Morten Sommer (msom@biosustain.dtu.dk)

Review Timeline:

Transfer Date:	10th Dec 24
Editorial Decision:	11th Dec 24
Revision Received:	12th Dec 24
Editorial Decision:	21st Jan 25
Revision Received:	25th Jan 25
Accepted:	29th Jan 25

Editor: Achim Breiling

Transaction Report: This manuscript was transferred to EMBO reports following peer review at Molecular Systems Biology.

Dear Prof. Sommer,

Thank you for transferring your manuscript to EMBO reports. I now went through the manuscript and through your point-by-point-response to the referee reports (attached again below). As you know, the referees at MSB had several concerns and suggestions to improve the manuscript, or to strengthen the data and the conclusions drawn. After reading your p-b-p-response, I would thus like to invite you to revise your manuscript, with the understanding that all concerns of the referees will be addressed in a revised manuscript or in a detailed p-b-p-response as indicated.

EMBO reports emphasizes novel functional over detailed mechanistic insight. Thus, we will not require addressing points regarding more mechanism experimentally. However, it will be necessary that during revision you address all points questioning the main conclusions of the study, and all technical concerns, or points regarding the experimental designs, model systems used, or data presentation.

Acceptance of your manuscript will depend on a positive outcome of another round of review at EMBO reports, using the same referees.

Revised manuscripts should be submitted within three months of a request for revision. Please contact me to discuss the revision if you have questions or comments regarding the revision, or should you need additional time.

- 1) a .docx formatted version of the final manuscript text (including legends for main figures, EV figures and tables), but without the figures included. Please make sure that changes are highlighted to be clearly visible. Figure legends should be compiled at the end of the manuscript text.
- 2) individual production quality figure files as .eps, .tif, .jpg (one file per figure), of main figures and EV figures. Please upload these as separate, individual files upon re-submission. Please make sure that all figure panels are called out separately and sequentially in the manuscript text

For more details please refer to our guide to authors:

See also our guide for figure preparation:

Moreover, please consult our guidelines for figure legend preparation:

- 4) a complete author checklist, which you can download from our author guidelines

(<https://www.embopress.org/page/journal/14693178/authorguide>). Please insert page numbers in the checklist to indicate where the requested information can be found in the manuscript. The completed author checklist will also be part of the RPF.

5) that primary datasets produced in this study (e.g. RNA-seq, ChIP-seq and array data) are deposited in an appropriate public database. This is now mandatory (like the COI statement). If no primary datasets have been deposited in any database, please state this in this section (e.g. 'No primary datasets have been generated and deposited').

The accession numbers and database should be listed in a formal "Data Availability " section (placed after Materials & Methods) that follows the model below. Please note that the Data Availability Section is restricted to new primary data that are part of this study.

Data availability

8) Regarding data quantification and statistics, please make sure that the number "n" for how many independent experiments were performed, their nature (biological versus technical replicates), the bars and error bars (e.g. SEM, SD) and the test used to calculate p-values is indicated in the respective figure legends (also for potential EV figures and all those in the final Appendix). Please also check that all the p-values are explained in the legend, and that these fit to those shown in the figure. Please provide statistical testing where applicable. Please avoid the phrase 'independent experiment', but clearly state if these were biological or technical replicates. Please also indicate (e.g. with n.s.) if testing was performed, but the differences are not significant. In case n=2, please show the data as separate datapoints without error bars and statistics.

See also:

<http://www.embopress.org/page/journal/14693178/authorguide#statisticalanalysis>

If n<5, please show single datapoints for diagrams. Please add to each legend (main, EV figures, Appendix, where applicable) a 'Data Information' section explaining the statistics used or providing information regarding replicates and scales. See: <https://www.embopress.org/page/journal/14693178/authorguide#figureformat>

9) Please add scale bars of similar style and thickness to any microscopic images, using clearly visible black or white bars (depending on the background). Please place these in the lower right corner of the images themselves. Please do not write on or near the bars in the image but define the size in the respective figure legend.

10) Please note our reference format:

11) We updated our journal's competing interests policy in January 2022 and request authors to consider both actual and perceived competing interests. Please review the policy <https://www.embopress.org/competing-interests> and add a statement declaring your competing interests. Please name that section 'Disclosure and Competing Interests Statement' and add it after the author contributions section.

12) Please add up to 5 keywords to the manuscript and order the sections like this using these names:
Title page - Abstract - Keywords - Introduction - Results - Discussion - Methods - Data availability section (DAS) -
Acknowledgements (including funding information) - Disclosure and Competing Interests Statement - References - Figure
legends - Expanded View Figure legends

13) Please provide the abstract written in present tense throughout and with not more than 175 words.

14) Please make sure that all the funding information is also entered into the online submission system and is complete and similar to the one in the manuscript text file (in the Acknowledgements).

15) We now use CRediT to specify the contributions of each author in the journal submission system. CRediT replaces the author contribution section. Please use the free text box to provide more detailed descriptions. Thus, please do NOT provide your final manuscript text file with an author contributions section. See also guide to authors:
<https://www.embopress.org/page/journal/14693178/authorguide#authorshipguidelines>

16) All materials and methods used need to be described in the main text using our 'Structured Methods' format, which is required for all research articles. According to this format, the Methods section should include a Reagents and Tools Table (listing key reagents, experimental models, software, and relevant equipment and including their sources and relevant identifiers), uploaded as separate file, followed by a Methods section in which we encourage the authors to describe their methods using a step-by-step protocol format with bullet points, to facilitate the adoption of the methodologies across labs. More information on how to adhere to this format as well as downloadable templates (.doc or .xls) for the Reagents and Tools Table can be found in our author guidelines (section 'Structured Methods'):

I look forward to seeing a revised version of your manuscript when it is ready. Please let me know if you have questions or comments regarding the revision.

Kind regards,

Achim

Referee #1:

In the manuscript titled "Modulating tumor immunity using advanced microbiome therapeutics producing an indole metabolite" Authors developed *Escherichia coli* Nissle 1917 that produce and secrete indole-3-acetic acid, aryl hydrocarbon receptor agonist, to promote bacteria-based cancer therapy via stimulation of T-cell immunity. The data could be consistent with a therapeutic effect, but the absence of appropriate control groups and several unclear sections, along with missing essential explanations, indicate that the study is likely incomplete. Overall, the manuscript does not adequately convey the ideas and results of the research. Additionally, key points have been overlooked in the discussion. Therefore, the manuscript is not deemed suitable for publication in *Molecular Systems Biology*.
The authors could enhance their manuscript by addressing the following comments.

Major:

1. The authors noted "A *tnsA* knockout (*tnaA*) EcN was generated ..." in the text (line 105-106). But it lacks supporting data to confirm. Moreover, since *E. coli* Nissle 1917 naturally expresses the *tnaA* gene, which converts tryptophan to indole and may activate aryl hydrocarbon receptor (AhR), the authors should evaluate the therapeutic effect of wild-type *E. coli* Nissle if they aim to demonstrate this mechanism.
2. The simultaneous expression of the three payloads (*aspC*, *ipdC*, *iad1*) under the control of the constitutive pMS6 promoter may affect the bacterial growth. Thus, bacterial growth should be evaluated.
3. What is the mechanism of secretion of the final product (IAA) out of bacteria?
4. In Fig. 1b, for the targeted high-resolution LC-HRMS analysis, the authors noted '2 μ L from each sample...'. It is unclear whether this volume refers to bacterial culture media, bacterial supernatant, or bacterial pellet.
5. In Fig. 1b and Table S1, the concentration of the microbial metabolite IAA produced by EcNIAA [780.14({plus minus}13.96) μ M] is significantly higher than that produced by EcNCtrl [5.08 ({plus minus}1.09) μ M] with a 153-fold. However, it is surprising, in Fig. 1c, that only 2-fold increase in AhR activity were observed in EcNIAA compared with EcNCtrl, which should be discussed.

Moreover, the AhR activity of the supernatant of EcNIAA [780.14(±13.96) μM] is lower than the positive control with 10 μM. This discrepancy needs to be addressed.

6. The AhR activity induced by EcNIAA and EcNCtrl is approximately 25,000 and 20,000 RLU, respectively (Fig. 1c).

Additionally, LD media also strongly induces AhR activity (10,000 RLU). This suggests that IAA does not play a major role in AhR activation."

7. Fig. 2-5 missed a control group. The in vivo study did not include a negative control group, such as PBS (isotonic saline). The authors had to do in vivo experiments including a PBS control group.

8. Although this study used the same mice background (BALB/c, C57BL/6) and cancer cell lines (CT26, MC38) for generating tumors, the variation in tumor sizes (75-200 mm³) is too large for bacterial treatment.

9. Authors noted "All animals harbored viable bacteria in tumors at the time of euthanization (Fig. 2d) up to 27 post-infection days, confirming the ability of EcN to colonize the tumors for longer periods". However, Fig. 2b shows rapid tumor growth at day 7, with 50% of the mice dying between days 15 and 27 after bacterial infection. This suggests that despite the prolonged tumor localization, the bacteria may have lost the pMUT1 plasmid responsible for producing IAA due to the absence of Kanamycin in vivo."

10. The test of plasmid loss should be specified both in vitro and in vivo.

11. The viable bacterial count in the other normal organs (spleens, lungs, kidneys, heart, blood) were not reported.

12. Flow cytometry analysis - the authors conducted the analysis on tumors but could have also analyzed other organs, such as the spleens and tumor-draining lymph nodes, from the sacrificed mice to monitor the immune cell subsets.

13. Bacteria possess innate immune-boosting properties against tumors due to intrinsic factors. Observation of innate immune cells (neutrophils, macrophages, NK cells) after bacterial treatment was not evaluated by FACS or H&E staining in this study.

14. The authors observed an increased number of total T-cells (Fig. 2f), but did not specify how many of these are Tregs, effector CD4⁺ T-cells, and activated CD8⁺ T cells by FACS analysis? Also, while they claim that T-cells have a tumor-suppressive effect, the literature has shown that their role is more complicated than this, with some subsets such as Treg (CD4⁺FOXP3⁺CD25⁺), effector CD4⁺ T-cells (CD4⁺FOXP3⁻), activated CD8⁺ T-cells (CD8⁺Gzmb⁺) associated with better prognosis in CRC. The authors could have investigated these T-cell subsets more thoroughly if they intend to support the claim that anti-tumor T-cell immunity against tumors is induced by EcNIAA.

15. Why did authors only focus on CD4⁺ and CD8⁺ T-cells by H&E staining but not include FACS analysis?

16. The assessment of systemic toxicity and body weight following bacterial treatment is essential and should be evaluated.

17. The absence of mention regarding mycoplasma testing in cancer cells is noted. This test is essential as mycoplasma can impact cancer cell proliferation.

Minor comments

1. The clone of FACS antibodies used in this study should be specified.

2. Live/dead staining should be used for in vivo FACS analysis.

3. Raw data of the main figures should be provided.

Referee #2:

The manuscript by Vaaben et al. describes the engineering of the probiotic *E. coli* Nissle strain to produce indole-3-acetic acid, an AHR agonist. The team demonstrates that the application of the engineered strain to CT26 subcutaneous tumors reduces tumor growth trajectory and improves mouse survival compared to the EcN strain that does not produce an AHR agonist. The authors provide evidence that CD4⁺ and CD8⁺ lymphocytes are enriched in the tumor environment treated with the engineered strain, and that IFN-γ, CXCL9, and IL-27 are up-regulated. The anti-tumor effects were also translatable to a second tumor model with MC38 cells and led to lasting effects. While this publication is well-written, my primary concern rests in the scientific novelty of the report. The engineered strain in EcN has been reported before and indole-3-acetic acid has been demonstrated to be produced by gut bacteria and reduce tumor burden in the same models used in this publication.

Specific comments:

Missing key references that engineered *E. coli* to secrete AHR agonists, including indole-3-acetic acid:

Kouno T. et al. Engineered bacteria producing aryl-hydrocarbon receptor agonists protect against ethanol-induced liver disease in mice. *Alcohol Clinical and Experimental Research*. 2023 (EcN)

Wu et al. High-Level Production of Indole-3-acetic Acid in the Metabolically Engineered *Escherichia coli*. *Biotechnology and Biological Transformations*. 2021 (*E. coli* MG1655)

Romasi et al. Development of indole-3-acetic acid-producing *Escherichia coli* by functional expression of *lpdC*, *AspC*, and *lad1*. *J Microbiol Biotechnol*. 2013

In the first paragraph of the results section, the engineered strain with *lpdC*, *AspC*, and *lad1* genes selected from a variety of microbial species needs to be referenced.

Include data from the different promoter mutants and the corresponding IAA production. Provide the sequences of the different RBS and combinations that resulted in the production of the 150-fold higher levels. This is arguably one of the key parts of the paper.

Figure 1A: After the addition of each gene, demonstrate that the expected product is formed.

Figure 1C: Should also include a dose-titration curve of the supernatants to enable the range of increased activation over the reporter assay. Additionally, a second means of confirming AHR reporter should be included since this reporter assay has been described to produce an AHR-independent signal dependent on SCFAs (Moudoux et al. *Gut Microbes*. 2022). qPCR for *cyp1a1*

or western blot is sufficient.

Line 121 "explicitly attributed" is too strong of a statement. Many indole derivatives activate AHR. Here a small number of tryptophan derivatives are measured. This does not meet the criteria for "explicitly attributed".

2F. Include FACS panels for other cell types beyond T-cells. Does the EcNIAA strain affect B, NK, macrophages, monocytes, etc?

Line 196. Minor typo > should be <

MC38 model should profile lymphocytes and IFN γ + cells

Referee #3:

In their manuscript, Vaaben et al. describe an engineered strain of *E. coli* Nissle for tumor therapy that produces indole-acetic-acid. They could show that such bacteria have improved anti-tumor activity. They could show that this is due to an alteration in the immigration of immune effector cells. This work is highly interesting and represents a further step towards application of this strategy to patients. The manuscript is well written, the work well designed and carried out (but see below).

Some comments, which still should be addressed:

1. It appears that the experiments were done only once. Reproducing of results is part of good laboratory practice.
2. Line 126: in what time did the bacteria produce this amount of IAA.
3. Line 282: mention that MC38 is syngeneic with C57Bl/6.

Dear Reviewers,

In preparing our revised manuscript we have taken the feedback from the reviewers and made substantial updates including several new datasets to the manuscript. In summary we have extended the data in the manuscript to include:

- **Strain validation and assessment:** Validation of the Δ tnaA knockout, evaluation of growth rates for the production strain, and comparison of AhR activity between wild-type and control strains.
- **In vitro experiments:** Expanded AhR activation analysis with a dose-response effect of IAA, as well as additional concentrations of spent medium from EcN^{IAA} to further validate the results.
- **In vivo experiments:** Expanded histological analysis in the CT26 model to assess the effects on innate immunity (macrophages/monocytes and neutrophils) and regulatory T-cells (FOXP3+) as well as adding bodyweight measurements to evaluate any systemic toxicity following administration.

We have provided a detailed point-by-point response to every concern raised by the reviewers.

With these new data and revised text, we believe that the manuscript is substantially strengthened and, we kindly ask that you reconsider our manuscript for review.

We thank the reviewers for their time and consideration, and we look forward to your response.

Kind regards,

Morten Sommer, PhD

Professor, Scientific Director

On behalf of the authors

Reviewer #1:

In the manuscript titled "Modulating tumor immunity using advanced microbiome therapeutics producing an indole metabolite" Authors developed Escherichia coli Nissle 1917 that produce and secrete indole-3-acetic acid, aryl hydrocarbon receptor agonist, to promote bacteria-based cancer therapy via stimulation of T-cell immunity. The data could be consistent with a therapeutic effect, but the absence of appropriate control groups and several unclear sections, along with missing essential explanations, indicate that the study is likely incomplete. Overall, the manuscript does not adequately convey the ideas and results of the research. Additionally, key points have been overlooked in the discussion. Therefore, the manuscript is not deemed suitable for publication in Molecular Systems Biology. The authors could enhance their manuscript by addressing the following comments.

1.1 1. The authors noted "A *tnaA* knockout (Δ *tnaA*) EcN was generated ..." in the text (line 105-106). But it lacks supporting data to confirm. Moreover, since *E. coli* Nissle 1917 naturally expresses the *tnaA* gene, which converts tryptophan to indole and may activate aryl hydrocarbon receptor (AhR), the authors should evaluate the therapeutic effect of wild-type *E. coli* Nissle if they aim to demonstrate this mechanism.

R: We appreciate the reviewer's comment regarding the need for data supporting the generation of the Δ *tnaA* knockout EcN strain. In response, we have added data to confirm the successful knockout, which is now included in **Figure EV2**. and added the following paragraph to the results section

*"The *tnaA* knockout was confirmed by PCR and gel electrophoresis (Fig. EV2a) followed by sanger sequencing".*

We also probed any changes in AhR activity between the wild-type and *tnaA* knockout strain and found no difference between the strains (Fig. S2d), and added the following paragraph in the main text

*"We tested the ability of the wild-type strain and the Δ *tnaA* knockout to activate AhR and found no difference (Fig. EV2d), further supporting the conclusion that AhR activation is driven by IAA in our engineered strain"*

1.2 The simultaneous expression of the three payloads (*aspC*, *ipdC*, *iad1*) under the control of the constitutive pMS6 promoter may affect the bacterial growth. Thus, bacterial growth should be evaluated.

R: We thank the reviewer for raising this point. To address this concern, we evaluated the effect of the Δ *tnaA* knockout as well as the introduction of the expression cassette containing *aspC*, *ipdC*, and *iad1* on bacterial growth performance. Our analysis revealed no notable differences in the growth of the engineered strains compared to controls, indicating that the simultaneous expression of the three payloads does not significantly impact bacterial growth. These results are now included in **Figures EV2b and EV2c**, and in the main text in the following paragraphs

*"We evaluated the effect of the Δ *tnaA* knockout and the introduction of the expression cassette containing *aspC*, *ipdC*, and *iad1* on the growth performance of the strains and observed no notable differences, indicating that the simultaneous expression of the three payloads did not significantly impact bacterial growth (Fig. EV2b and EV2c)"*

1.3 What is the mechanism of secretion of the final product (IAA) out of bacteria?

R: We thank the reviewer for this comment. Much is still unknown about bacterial transport systems, and a detailed investigation of the IAA secretion mechanism is outside the scope of this paper. Importantly, we have demonstrated that IAA is secreted by quantifying its presence in the spent medium (Fig. 1b), confirming that it is indeed released by the engineered strain. We hypothesize that the transport occurs through the native TolC transporter system or through simple diffusion across the membrane. We added the following paragraph in the main text to clarify on this matter:

“These results demonstrate that we have successfully engineered a strain capable of secreting high quantities of IAA without compromising bacterial fitness. The mechanism of IAA transport outside the bacteria remains unknown, but it may occur through native efflux pumps, or through passive diffusion across the lipid membrane”

1.4 In Fig. 1b, for the targeted high-resolution LC-HRMS analysis, the authors noted '2 μ L from each sample...'. It is unclear whether this volume refers to bacterial culture media, bacterial supernatant, or bacterial pellet.

R: We apologize for this lack of clarity in the manuscript and have updated it accordingly:

"For analysis, 2 μ L from each sample (spent medium from bacterial fermentation, or tumor homogenate) was introduced into a high-efficiency liquid chromatography quadrupole time-of-flight mass spectrometry system..."

1.5 In Fig. 1b and Table S1, the concentration of the microbial metabolite IAA produced by EcNIAA [780.14({plus minus}13.96) μ M] is significantly higher than that produced by EcNCtrl [5.08 ({plus minus}1.09) μ M] with a 153-fold. However, it is surprising, in Fig. 1c, that only 2-fold increase in AhR activity were observed in EcNIAA compared with EcNCtrl, which should be discussed. Moreover, the AhR activity of the supernatant of EcNIAA [780.14({plus minus}13.96) μ M] is lower than the positive control with 10 μ M. This discrepancy needs to be addressed.

R: We appreciate the reviewer's observation regarding the apparent discrepancy between IAA concentration and AhR activity. It is important to note that only 10% of the bacterial supernatant is added to the reporter cells, which results in a dilution of the IAA concentration in the assay. Additionally, we believe the observed lower-than-expected AhR activation may be influenced by baseline activation due to the presence of alternative compounds or other factors in the rich LB medium. This background signal can dampen the relative increase in AhR activation induced by IAA. To address this concern, we repeated the assay using different volumes of spent medium and observed a dose-dependent increase in AhR activation, further supporting the role of IAA in driving AhR signaling. Some biological variation and batch-to-batch differences between the fermentations used for LC/MS and the cell-based assay could also contribute to the observed differences in AhR activation. Nonetheless, IAA remains an important driver of the AhR activation observed in these experiments. We have added the following paragraph in the main text to address the difference between activation with pure compound and the spent medium:

"The discrepancy between the results observed when exposing cells to pure IAA versus the spent medium in AhR activity may be attributed to the presence of other substances in the spent medium, such as LPS, which could induce stress responses. Additionally, we observed that using more than 20% spent medium caused a color change in the medium, likely due to a shift in pH, which may have further influenced the AhR activity response."

1.6 The AhR activity induced by EcN^{IAA} and EcN^{Ctrl} is approximately 25,000 and 20,000 RLU, respectively (Fig. 1c). Additionally, LD media also strongly induces AhR activity (10,000 RLU). This suggests that IAA does not play a major role in AhR activation."

R: We thank the reviewer for pointing this out. To address this comment, we repeated the AhR activity assay with different concentrations of spent medium from EcN^{IAA} (as also suggested by another reviewer), as shown in **Figure 1e**. Additionally, we ensured that the medium was not frozen before conducting the assay this time, to avoid potential artifacts caused by freezing and thawing. The updated results confirm that while baseline activation from LB medium contributes to the observed signal, IAA continues to play a significant role in AhR activation. We have also added the following sentence to the revised manuscript to address the discrepancy, as requested:

"The discrepancy between the results observed when exposing cells to pure IAA versus the spent medium in AhR activity may be attributed to the presence of other substances in the spent medium, such as SCFA or LPS, which could induce stress responses. Additionally, we observed that using more than 20% spent medium caused a color change in the medium, likely due to a shift in pH, which may have further influenced the AhR activity response."

1.7 Fig. 2-5 missed a control group. The in vivo study did not include a negative control group, such as PBS (isotonic saline). The authors had to do in vivo experiments including a PBS control group.

R: We appreciate the reviewer's suggestion; however, we believe that including a PBS control group in this study would not directly address the primary research question. Our focus is on the specific therapeutic potential of the engineered *Escherichia coli Nissle* strain and its ability to produce indole-3-acetic acid (IAA) in the context of cancer treatment. As we are not investigating the natural anti-cancer effects of *E. coli Nissle* itself, we feel that a PBS group would not provide additional relevant insights. Instead, we selected the Δ tnaA knockout strain with an empty expression cassette as the most appropriate control, as it enables us to specifically evaluate the contribution of IAA production to the observed anti-tumor effects. We have also previously worked with the CT-26 model and observed that *E. coli Nissle* itself does not have an anti-tumor effect when compared to PBS (<https://www.nature.com/articles/s41598-023-39365-2>, Figure 3). In alignment with the principles of minimizing animal use we made the active decision not to include a PBS group.

1.8 Although this study used the same mice background (BALB/c, C57BL/6) and cancer cell lines (CT26, MC38) for generating tumors, the variation in tumor sizes (75-200 mm³) is too large for bacterial treatment.

R: We respectfully disagree with the reviewer's concern regarding tumor size variability. The size threshold for starting therapy often varies across studies and is necessarily somewhat arbitrary. Our chosen range falls well within what is considered standard in similar research, as demonstrated in studies like the one by Leventhal et al., 2020 (*Nature Communications*), where "tumors were allowed to establish until they reached between 40 and 300 mm³" (<https://doi.org/10.1038/s41467-020-16602-0>). Including a more heterogeneous range of tumor sizes ensures that our findings are not limited to a specific subtype, such as slower-growing tumors, and instead reflect the variability and heterogeneity inherent to cancer in the real world. We believe this approach enhances the relevance and generalizability of our results while maintaining their validity.

1.9 Authors noted "All animals harbored viable bacteria in tumors at the time of euthanization (Fig. 2d) up to 27 post-infection days, confirming the ability of EcN to colonize the tumors for longer periods". However, Fig. 2b shows rapid tumor growth at day 7, with 50% of the mice dying between days 15 and 27 after bacterial infection. This suggests that despite the prolonged tumor localization, the bacteria may have lost the pMUT1 plasmid responsible for producing IAA due to the absence of Kanamycin in vivo."

R: We thank the reviewer for this comment. All our experiments were conducted using EcN strains harbouring the pMUT1 plasmid, which includes a kanamycin resistance cassette. While Kanamycin was used for selection during strain construction, it was not necessary to include in the in vivo experiments, as plasmid stability was ensured by the Hok/Sok toxin-antitoxin system encoded on the plasmid (Figure EV1). Additionally, throughout the experiments, bacterial colonies recovered from tumors retained the plasmid, as confirmed by growth on selective media containing Kanamycin, further supporting the stability of the plasmid in the absence of antibiotic pressure (<https://pubmed.ncbi.nlm.nih.gov/34842418/>). We have updated the following section in the main text to clarify on this matter:

"The presence of viable bacteria in tumors was determined by plating serial dilutions of tumor homogenates on plates containing kanamycin. All animals harbored viable bacteria in tumors at the time of euthanization (Fig. 2c), confirming the ability of EcN to colonize the tumors for longer periods and that the pMUT1 based plasmid is also retained."

Furthermore, the observed "exponential" growth of tumors aligns with the typical tumor progression seen in murine models (<https://aacrjournals.org/cancerrescommun/article/4/8/2267/747422/Mathematical-Modeling-of-Tumor-Growth-in>), and we have no evidence to suggest that this was related to plasmid loss. We believe this growth pattern is due to the natural course of tumor progression rather than any loss of plasmid or therapeutic efficacy.

1.10 The test of plasmid loss should be specified both in vitro and in vivo.

R: We appreciate the reviewer's suggestion but testing for plasmid loss in vitro and in vivo falls outside the scope of this study. The pMUT1 plasmid has been demonstrated to remain stable in *Escherichia coli* Nissle 1917 in previous studies

(DOI: [10.1021/acssynbio.0c00466](https://doi.org/10.1021/acssynbio.0c00466)), which is further supported by the inclusion of the Hok/Sok toxin-antitoxin system designed to enhance plasmid stability. Additionally, we verified plasmid retention by plating bacteria recovered from tumors on selective media containing Kanamycin after in vivo experiments. This approach confirms the stable presence of the plasmid throughout the experimental timeline without the need for further plasmid stability testing in this specific study. As mentioned in the comment above, we have revised the main text to better clarify the use of kanamycin on plates used to compute the CFU:

“The presence of viable bacteria in tumors was determined by plating serial dilutions of tumor homogenates on plates containing kanamycin. All animals harbored viable bacteria in tumors at the time of euthanization (Fig. 2c), confirming the ability of EcN to colonize the tumors for longer periods and that the pMUT1-based plasmid remains stable over extended periods, even in the absence of antibiotic selective pressure.”

1.11 The viable bacterial count in the other normal organs (spleens, lungs, kidneys, heart, blood) were not reported.

R: We thank the reviewer for this comment. In our study we chose to focus on the liver for bacterial colonization analysis because it is the organ most associated with off-target colonization in our model, as well as in previous studies using *Escherichia coli* Nissle 1917 (EcN) strains. Our previous experiments, as well as the broader literature, consistently demonstrate that the liver is the primary site of off-target colonization, making it the most relevant organ for assessing the safety and specificity of tumor colonization by EcN. As such, we did not assess viable bacterial counts in other organs in this study. We refer to the following sentence in the main text:

“We also investigated potential off-target colonization and found no viable bacteria in livers, the organ most associated with off-target colonization^{28,29} at the time of euthanization (Fig. 2e).”

1.12 Flow cytometry analysis - the authors conducted the analysis on tumors but could have also analyzed other organs, such as the spleens and tumor-draining lymph nodes, from the sacrificed mice to monitor the immune cell subsets.

R: We agree that analysing immune cell subsets in organs such as the spleens and tumor-draining lymph nodes could provide additional insights, but unfortunately, we did not collect or save these tissues for this study. Our hypothesis focused on the local immune response and changes within the tumor microenvironment, which is why our analysis was limited to the tumors. We acknowledge that future studies could benefit from a broader analysis of immune cell subsets in peripheral organs to complement the findings observed in the tumor.

1.13 Bacteria possess innate immune-boosting properties against tumors due to intrinsic factors. Observation of innate immune cells (neutrophils, macrophages, NK cells) after bacterial treatment was not evaluated by FACS or H&E staining in this study.

R: We thank the reviewer for the comment. To address this concern, we extended the histological analysis to include a evaluation of myeloid cells, specifically macrophages and neutrophils, to address the potential role of innate immune cells in the tumor microenvironment. Using immunohistochemistry, we analyzed CD68 as a

marker for macrophages/monocytes and ELA2 (neutrophil elastase) for neutrophils. These findings are presented in Figure 3 and discussed in the results section in the updated version of the manuscript.

“We also evaluated the effect on myeloid cells using CD68 as a marker for macrophages and monocytes, and ELA2 (neutrophil elastase) as a marker for neutrophils.... We observed no significant differences in the number of CD68+ macrophages/monocytes or ELA2+ neutrophils between the treatment and control groups, suggesting that EcN^{IAA} treatment primarily influences adaptive immune cells rather than myeloid cell populations.”

1.14 The authors observed an increased number of total T-cells (Fig. 2f), but did not specify how many of these are Tregs, effector CD4+ T-cells, and activated CD8+ T cells by FACs analysis? Also, while they claim that T-cells have a tumor-suppressive effect, the literature has shown that their role is more complicated than this, with some subsets such as Treg (CD4+FOXP3+CD25+), effector CD4+ T-cells (CD4+FOXP3-), activated CD8+ T-cells (CD8+GzmB+) associated with better prognosis in CRC. The authors could have investigated these T-cell subsets more thoroughly if they intend to support the claim that anti-tumor T-cell immunity against tumors is induced by EcN^{IAA}

R: We thank the reviewer for this comment. We initially aimed to conduct a more detailed subset analysis using flow-cytometry, but the viability of the tissue was too low to provide reliable results. This limitation is discussed in the manuscript: *“Due to reduced cell viability from cryopreservation, cell counts within CD4+ and CD8+ populations were insufficient (>500 cells) for meaningful analysis of downstream subsets.”* Therefore, we were unable to provide a more granular breakdown of the T-cell populations beyond the total CD3+ cells.

The reviewer is correct that the role of T-cells in the tumor microenvironment is complex, and we acknowledge this. However, as the first step in any adaptive immune response, it is critical that immune cells, particularly T-cells, infiltrate the tumor in sufficient numbers, which is widely recognized in the context of “hot” versus “cold” tumors. Thus, the total immune cell infiltration is a key readout on its own, serving as an initial indicator of immune activation within the tumor. Furthermore, we have included granzyme B staining in our analysis to detect cytotoxic cells, such as NK and activated T-cells, providing additional insight into the functional aspects of the immune response. Lastly, we now extended the histology analysis to include FOXP3-positive cells (Tregs), and we observed that the levels of Tregs are not different between the groups, entailing that the ratio of CD4⁺ helper T cells (Th) to regulatory T cells (Tregs) is shifted. This shift suggests that EcN^{IAA} treatment promotes a more pro-inflammatory adaptive immune response, rebalancing the tumor microenvironment towards a state more conducive to anti-tumor immunity. The following sentences have been added to the results:

“Interestingly, as the percentage of FOXP3⁺ regulatory T cells remained unchanged between the groups, the data suggest that EcN^{IAA} treatment shifts the ratio of CD4⁺ helper T cells to regulatory T cells (Tregs), favoring a more pro-inflammatory adaptive immune response.”

And the discussion:

“Higher levels of CD8+ lymphocyte infiltration in tumors have been consistently associated with improved overall survival in multiple cohorts of CRC patients”

1.15 Why did authors only focus on CD4+ and CD8+ T-cells by H&E staining but not include FACS analysis?

R: We thank the reviewer for this question. To clarify, we do have flow-cytometry data on total CD3+ T cells, as presented in Figure 2g. Unfortunately, due to the low viability of the frozen tumor samples, we were unable to obtain reliable results from any downstream subsets, as mentioned in the manuscript. However, we believe that our use of single-cell marker specificity through whole-slide immunohistochemistry provides a robust and quantitative readout. Although we recognize the value of flow-cytometry analysis for additional subset resolution, we believe our approach offers a strong quantitative and spatial assessment of immune cell infiltration.

1.16 The assessment of systemic toxicity and body weight following bacterial treatment is essential and should be evaluated.

R: We thank the reviewer for this suggestion and have now included the assessment of bodyweight in our revised manuscript. Bodyweight trajectories following bacterial treatment have been added to the results and are shown in **Figures 2d** and **5d**. These data demonstrate that there were no significant changes in bodyweight, nor any differences between the treatment groups, indicating that the bacterial treatment was well-tolerated and did not induce systemic toxicity. We have added the following paragraph in the main text to reflect this change:

“The administration of bacteria was well tolerated in all animals, with no significant decrease in bodyweight in either group (Fig. 2d).”

1.17 The absence of mention regarding mycoplasma testing in cancer cells is noted. This test is essential as mycoplasma can impact cancer cell proliferation.

R: We thank the reviewer for pointing out this omission and have now added the relevant information to the Materials and Methods section. All cell lines used for the animal experiments were initially tested for the presence of mycoplasma to ensure their integrity and suitability for experimental use. The following section has been added in the materials and methods:

“All cell lines were tested for the presence of mycoplasma to ensure their integrity and suitability for experimental use.”

Minor comments:

1.18 The clone of FACS antibodies used in this study should be specified.

R: We thank the reviewer for this suggestion and have now added the clone numbers for all FACS antibodies used in this study to **Table 2**. For the **eFluor 780 viability**

dye, no clone number is applicable, as it is a viability dye rather than an antibody. The catalog number has been provided for reference.

Table 1 – Antibodies used in T-cell panel (Target – Color, Supplier, Clone number)

CD45-BUV395, #564279 BD, Clone 30-F11
IFNγ-BV421, #563376 BD, Clone XMG1.2
CD4-FITC, #553047 BD, Clone RM4-5
GzmB-PE, #561142 BD, Clone GB11
CD3-PECF594, #562332 BD, Clone 145-2C11
CD8-APC, #553035 BD, Clone 53-6.7
Dead cells – eFluor 780, #65-0865-18, Thermo

1.19 Live/dead staining should be used for in vivo FACS analysis

R: We thank the reviewer for their comment. Live/dead staining was indeed performed for all FACS analyses. We have updated the Materials and Methods section to make this clearer:

“Only viable cells were included in the analysis, as determined by gating based on staining with the viability dye eFluor™ 780 (Thermo Fisher Scientific catalog number 65-0865-18).”

The gating strategy, including the use of viability dye, is also detailed in **Figure EV4**.

1.20 Raw data of the main figures should be provided.

R: We thank the reviewer for the comment and have added the raw data used to generate the figures as an excel file.

Reviewer #2:

The manuscript by Vaaben et al. describes the engineering of the probiotic E. coli Nissle strain to produce indole-3-acetic acid, an AHR agonist. The team demonstrates that the application of the engineered strain to CT26 subcutaneous tumors reduces tumor growth trajectory and improves mouse survival compared to the EcN strain that does not produce an AHR agonist. The authors provide evidence that CD4+ and CD8+ lymphocytes are enriched in the tumor environment treated with the engineered strain, and that IFN- γ , CXCL9, and IL-27 are up-regulated. The anti-tumor effects were also translatable to a second tumor model with MC38 cells and led to lasting effects. While this publication is well-written, my primary concern rests in the scientific novelty of the report. The engineered strain in EcN has been reported before and indole-3-acetic acid has been demonstrated to be produced by gut bacteria and reduce tumor burden in the same models used in this publication.

R: We would like to thank the reviewer for their thoughtful comments and for recognizing the merit of our work. We appreciate the opportunity to address the concern regarding the scientific novelty of our report. While it is true that the production of indole-3-acetic acid (IAA) by gut bacteria has been reported before, we are not aware of any previous publications that demonstrate the effects we have observed in our study. Specifically, the modulation of tumor immunity and the observed changes in tumor microenvironment through the application of engineered *E. coli* Nissle producing IAA have not, to our knowledge, been reported.

In our view the most relevant prior study we are aware of that investigates the therapeutic potential of IAA in cancer is the work by Tintelnot et al., 2023 (*Bacterial 3-IAA enhances efficacy of chemotherapy in pancreatic cancer*, DOI: [10.1016/j.xcrm.2023.101039](https://doi.org/10.1016/j.xcrm.2023.101039)). However, this study focuses on the role of IAA in potentiating the anticancer effect of chemotherapy against pancreatic ductal adenocarcinoma, rather than its direct application in modulating tumor immunity as we have demonstrated in our syngeneic models.

We hope this clarifies the novelty of our findings, and we are grateful for the reviewer's feedback.

2.1 Missing key references that engineered *E. coli* to secrete AHR agonists, including indole-3-acetic acid:

Kouno T. et al. Engineered bacteria producing aryl-hydrocarbon receptor agonists protect against ethanol-induced liver disease in mice. *Alcohol Clinical and Experimental Research*. 2023 (EcN)

Wu et al. High-Level Production of Indole-3-acetic Acid in the Metabolically Engineered *Escherichia coli*. *Biotechnology and Biological Transformations*. 2021 (*E. coli* MG1655)

Romasi et al. Development of indole-3-acetic acid-producing *Escherichia coli* by functional expression of *lpdC*, *aspC*, and *lad1*. *J Microbiol Biotechnol*. 2013

In the first paragraph of the results section, the engineered strain with *lpdC*, *aspC*, and *lad1* genes selected from a variety of microbial species needs to be referenced.

R: We thank the reviewer for their thoughtful comment. We have now referenced the previous papers describing the biosynthesis of IAA using the pathway we employed in the first results section (Romasi et al., 2013). Additionally, we have included a discussion of previous research producing IAA in situ for therapeutic purposes in the Discussion section with the following sentence,

“While E. coli has previously been engineered to produce IAA in situ^{51,52}, we significantly increased the production titres compared to earlier studies and focused on production using the probiotic E. coli Nissle 1917 strain, which is compatible with health-related applications. Our work focuses not only on significantly enhancing IAA production in EcN, but also on its application as an anticancer therapy, demonstrating the role of IAA in modulating tumor immunity and suppressing tumor growth.”

We hope these additions clarify the context of our work and highlight its novelty in the field of cancer therapy.

2.2 Include data from the different promoter mutants and the corresponding IAA production. Provide the sequences of the different RBS and combinations that resulted in the production of the 150-fold higher levels. This is arguably one of the key parts of the paper.

R: We appreciate the reviewer's comment. We have updated **Figure EV1** with the sequences, including the names of the RBS (RBS3, RBS4) used in the design of the constructs. These sequences are now clearly indicated under the corresponding parts. In addition, the strain names have been specified in **Table 1**, such as the MS6-RBS4-aspC-RBS4-ipdC-RBS3-iaa1 (EcN^{IAA}) strain. This information ensures full reproducibility of the constructs used in this study.

We would like to clarify that the reported 150-fold increase in IAA production refers to the comparison between the IAA-producing strain (EcN^{IAA}) and the non-expressing control strain. The elevated production titers are the result of the combination of the plasmid backbone, the strong promoter previously developed in our lab, and the tested RBS combinations, all of which contribute to the observed increase in IAA levels.

2.3 Figure 1A: After the addition of each gene, demonstrate that the expected product is formed.

R: We thank the reviewer for this comment. In the previous version of the figure, we aimed to indicate the formation of the expected products by using arrows that follow the color coding of the genes. However, to improve clarity, we have now tweaked the visuals of **Figure 1A**, making the gene-product relationships more distinct. Additionally, we have added an illustration of the final product, IAA, being secreted from the cell to further highlight the completion of the biosynthetic pathway.

2.4 Figure 1C: Should also include a dose-titration curve of the supernatants to enable the range of increased activation over the reporter assay. Additionally, a second means of confirming AHR reporter should be included since this reporter assay has been described to produce an AHR-independent signal dependent on SCFAs (Moudoux et al. Gut Microbes. 2022). qPCR for *cyp1a1* or western blot is sufficient.

R: We thank the reviewer for this suggestion. We have now included a dose-titration curve of the supernatants in **Figure 1e**, allowing for a clearer view of the range of increased AhR activation over the reporter assay.

Regarding the inclusion of a second method to confirm AhR activity, the referenced paper (Moudoux et al., 2022) discusses how SCFAs, like butyrate, modulate AhR signalling indirectly through HDAC inhibition, which leads to epigenetic changes. Since this modulation affects downstream gene expression, including assays such as qPCR for *cyp1a1*, any additional assay would still be subject to the same regulation.

Additionally, the reporter cell line we use in our study contains the entire regulatory sequence of the *Cyp1a1* gene, meaning that any activation we observe in this system is already a reflection of *Cyp1a1* gene regulation. Therefore, performing a separate qPCR or western blot for *Cyp1a1* would essentially be redundant, as it would measure the same downstream effect as the reporter system.

We believe that by comparing the AhR activity of EcN^{IAA} to the control strain (EcN^{Ctrl}), which are expected to produce similar amounts of SCFA, we can specifically

demonstrate the contribution of IAA to AhR activation while accounting for potential SCFA-induced effects.

We hope the reviewer accepts the extended explanation of our rationale and agrees with our approach.

2.5 Line 121 "explicitly attributed" is too strong of a statement. Many indole derivatives activate AHR. Here a small number of tryptophan derivatives are measured. This does not meet the criteria for "explicitly attributed".

R: We appreciate the reviewer's feedback and agree that "explicitly attributed" may have been too strong a phrasing. In response, we have removed the word "explicitly" from the manuscript to better reflect the contribution of indole derivatives to AhR activation, as measured in our experiments.

2.6 2F. Include FACS panels for other cell types beyond T-cells. Does the EcNIAA strain affect B, NK, macrophages, monocytes, etc?

We appreciate the reviewer's suggestion. While it would be ideal to examine a broader range of immune cell types, flow cytometry is inherently limited by the number of lasers and detectors available on a given instrument, which restricts the number of immune cell subsets that can be analyzed simultaneously. We used a flow cytometer with multiple lasers, allowing for considerable multiplexing, but due to these experimental constraints, we had to prioritize specific immune cell populations of interest. A more comprehensive approach, such as single-cell RNA sequencing, could provide additional insights, though this method is unfortunately cost-prohibitive.

However, in response to feedback from both the reviewer and another reviewer, we have expanded our histology analysis to include innate immune cells, specifically macrophages and neutrophils cells, this additional data can be found in the updated version of figure 3, and is discussed in the following paragraph of the results:

"We also evaluated the effect on myeloid cells using CD68 as a marker for macrophages and monocytes, and ELA2 (neutrophil elastase) as a marker for neutrophils... We observed no significant differences in the number of CD68+ macrophages/monocytes or ELA2+ neutrophils between the treatment and control groups, suggesting that EcN^{IAA} treatment primarily influences adaptive immune cells rather than myeloid cell populations.

2.7 Line 196. Minor typo > should be <'

R: Thank you for pointing this out. We have corrected the typo.

2.8 MC38 model should profile lymphocytes and IFN γ + cells

R: We thank the reviewer for this valuable suggestion. While we agree that profiling lymphocytes and IFN- γ + cells in the MC38 model would provide additional insights, we chose to focus our detailed immune profiling on the CT26 model due to resource

limitations. The CT26 model was selected as our primary system for immune analysis, and the data generated from this model provided robust and informative results regarding the immune response to our engineered strain.

While we recognize the value of extending this analysis to the MC38 model, practical limitations in resources and capacity prevented us from including it in the current study. We believe the insights gained from the CT26 model still provide valuable information about the role of IAA in modulating tumor immunity, and we hope to explore this further in future work.

Reviewer #3:

In their manuscript, Vaaben et al. describe an engineered strain of *E. coli* Nissle for tumor therapy that produces indole-acetic-acid. They could show that such bacteria have improved anti-tumor activity. They could show that this is due to an alteration in the immigration of immune effector cells. This work is highly interesting and represents a further step towards application of this strategy to patients. The manuscript is well written, the work well designed and carried out (but see below). Some comments, which still should be addressed: 1. It appears that the experiments were done only once. Reproducing of results is part of good laboratory practice. 2. Line 126: in what time did the bacteria produce this amount of IAA. 3. Line 282: mention that MC38 is syngeneic with C57Bl/6.

R: We are pleased that the reviewer found our work exciting and appreciate their positive feedback on the design and execution of our study. We are happy to address the specific comments raised and have provided detailed responses below to clarify and expand on the points mentioned. We hope that our revisions further strengthen the manuscript and address any outstanding concerns.

3.1 The experiments appear to have been conducted only once, and results should be reproduced for validation.

R: We appreciate the reviewer's comment, although we would like to clarify that the *in vitro* experiments were conducted multiple times to ensure reproducibility. In particular, we have now included a revised figure displaying AhR activity and a dose-response curve in **Figure 1**, further validating and reproducing our findings.

For the *in vivo* studies, we demonstrated efficacy in two separate syngeneic tumor models (CT26 and MC38), which not only reproduces the results but also extends them to a broader biological context. This approach allows us to demonstrate the robustness and generalizability of our findings across different tumor models.

3.2 Clarification needed on the time frame in which bacteria produced the observed amount of IAA (line 126).

R: We thank the reviewer for their comment. We have updated both the main text and the Materials and Methods section to explicitly state that the observed amount of IAA was produced after 24 hours of fermentation.

“Quantitative analysis of tryptophan-derived metabolites was conducted on the spent medium from the engineered strains and the control strain harboring the empty pMUT1 (EcN^{Ctrl}) vector (Fig. 1b, Table 1) after 24h fermentation”

3.3 Line 282 should mention that MC38 is syngeneic with C57Bl/6.

R: We thank the reviewer for their suggestion. We have now added clarification in the manuscript, stating that the MC38 model is syngeneic to C57Bl/6 mice.

“To assess if the effects observed in the CT26 model could be translated to other animal models of CRC, we tested the effect of EcN^{IAA} in the MC38 immunocompetent model, syngeneic to C57Bl/6 mice.”

Dear Dr. Sommer,

Thank you for the submission of your revised manuscript to our editorial offices. I have now received the reports from two of the three referees that I asked to re-evaluate the study, you will find below. As you will see, the two referees now fully support the publication of the study in EMBO reports. Original referee #1 was completely unresponsive to my invitations to re-assess the study. However, going through your p-b-p-response, I consider the points of this referee as adequately addressed.

Present referee #2 has suggestions to improve the manuscript I ask you to address in a final revised manuscript. Moreover, please have the final manuscript carefully proofread by a native speaker, as suggested also by referee #2.

- We plan to publish your manuscript as Report, as there are only 5 main figures and 4 EV figures. For a Scientific Report we require that results and discussion sections are combined in a single chapter called "Results & Discussion". Please do this for your manuscript. For more details please refer to our guide to authors:
<http://www.embopress.org/page/journal/14693178/authorguide#researcharticleguide>
- Please order the manuscript sections like this, using these names:
Title page - Abstract - Keywords - Introduction - Results & Discussion - Methods - Data availability section - Acknowledgements - Disclosure and Competing Interests Statement - References - Figure legends - Expanded View Figure legends
- Please remove the part "ORCID for corresponding author" from the manuscript text file. The ORCID will be directly linked to the corresponding author in the online version of the paper.
- Please make sure that all figure panels (main and EV figures) are called out separately and sequentially. Presently, there seem to be no separate callouts for panels 2G, 3B, 5A and 5F. Please check.
- Please check again that the number "n" for how many independent experiments were performed, their nature (biological versus technical replicates), the bars and error bars (e.g. SEM, SD) and the test used to calculate p-values is indicated in the respective figure legends (main and EV figures). Please also check that all the p-values are explained in the legend, and that these fit to those shown in the figure. Please provide statistical testing where applicable. Please avoid the phrase 'independent experiment', but clearly state if these were biological or technical replicates. Please also indicate (e.g. with n.s.) if testing was performed, but the differences are not significant. In case n=2, please show the data as separate datapoints without error bars and statistics. See also:
<http://www.embopress.org/page/journal/14693178/authorguide#statisticalanalysis>
- If n<5, please show single datapoints for diagrams. Presently, it seems 'n.s.' is not indicated in several diagrams. Moreover:
 - Please provide information related to n in the legends of figures 2B, D, F, G; 4A, B; 5B, D, F.
 - Please note that n=2 in figures 1B. Thus, please show the two datasets separately and remove the statistics.
 - Please define the error bars in the legends of figures EV2 B-D.
 - Please note that the figure EV2 subfigure "D" is mislabeled as figure "C" in the manuscript. This needs to be rectified.
 - Please provide exact p values in the legends of figures 1B, D, E; 2B, C, G; 3C; 4A, B; 5B, C, G.
- Please move the information shown in Table 2 to the Reagents and Tools table and update the callout. Then please remove Table 2 from the manuscript text file. Moreover, please add callouts to the reagents and tools table in the methods section where applicable.
- Please add scale bars of similar style and thickness to microscopic images, using clearly visible black or white bars (depending on the background). Please place these in the lower right corner of the images themselves. Please do not write on or near the bars in the image but define the size in the respective figure legend.
- Please use our reference format ('et al' needs to be used after 10 author names; DOIs should only be used for preprints and datasets that have not been published yet):
<http://www.embopress.org/page/journal/14693178/authorguide#referencesformat>
- The should the light microscopic images (left column) and the IF images (right column) in Fig. EV3 show the same tumor image for each row? I seems for some images this is not the case. Please check.
- For the source data for Fig. 3B please provide a TIFF file of the actual images shown, as suggested.

In addition, I would need from you uploaded separately:

- a short, two-sentence summary of the manuscript (not more than 35 words).

- two to four short (!) bullet points highlighting the key findings of your study (two lines each).
- a schematic summary figure as separate file that provides a sketch of the major findings (not a data image) in jpeg or tiff format (with the exact width of 550 pixels and a height of not more than 400 pixels) that can be used as a visual synopsis on our website.

Best,

Referee #1 (original referee #2):

The manuscript by Vaaben et al. provides an in-depth exploration of the engineering of the probiotic *E. coli* Nissle strain to synthesize indole-3-acetic acid, a known agonist of the aryl hydrocarbon receptor (AHR). This approach leverages the potential of engineered probiotics to modulate immune responses, aiming to reduce tumor growth. In their resubmission, the authors have made significant enhancements to their original manuscript. They have addressed my earlier concerns by including additional experimental evidence, controls, clearer explanations of their methodologies and results, and appropriate references. These improvements considerably strengthen the manuscript, contributing to its compelling biological story.

Referee #2 (original referee #3):

The revised manuscript of Vaaben et al is now suitable for publication. All the points that have been raised are appropriately dealt with. Recently, a certain hype on *E. coli* Nissle as therapeutic agent against tumors could be observed. However, Nissle has been shown to be inferior in this respect compared to attenuated bacterial pathogens. Therefore, it is a very logical effort to arm such probiotic bacteria with additional properties as the authors have done. Therefore, I find this work very important. It should be of interest for a general audience. It is unfortunate that the authors did not test their strain by systemic administration. The data are clearly presented and in general, the manuscript is well written. At some places, a native speaker could improve the style.

A few very minor points might render the manuscript even easier to understand:

1. I would suggest to add the Kana resistance cassette and the Hoc/Soc cassette into the schematic drawing of the plasmid in Figure 1a already, to make clear how the plasmid was maintained. These two features should be also mentioned in M&M and ev. Results.
2. In the Introduction line 84/85 the recent publication by Redenti et al. Nature November 14, 2024 should be added.
3. The time at which measurements were done should be mentioned for the culture as well as for the ex vivo analysis.

Dear Achim Breiling,

We are grateful to the editor and the reviewers for the efficient process. We are pleased to hear that both referees found the updated manuscript to have improved considerably. We will provide a brief point-by-point response to the minor points raised by referee #2, as well as the editorial checks.

Referee #2 (original referee #3):

The revised manuscript of Vaaben et al is now suitable for publication. All the points that have been raised are appropriately dealt with. Recently, a certain hype on *E. coli* Nissle as therapeutic agent against tumors could be observed. However, Nissle has been shown to be inferior in this respect compared to attenuated bacterial pathogens. Therefore, it is a very logical effort to arm such probiotic bacteria with additional properties as the authors have done. Therefore, I find this work very important. It should be of interest for a general audience. It is unfortunate that the authors did not test their strain by systemic administration. The data are clearly presented and in general, the manuscript is well written. At some places, a native speaker could improve the style.

A few very minor points might render the manuscript even easier to understand:

1. I would suggest to add the Kana resistance cassette and the Hoc/Soc cassette into the schematic drawing of the plasmid in Figure 1a already, to make clear how the plasmid was maintained. These two features should be also mentioned in M&M and ev. Results.

R: We have updated figure 1A to include these components, and the materials and methods as well as the plasmid map in figure EV1 contains the sequence of all genetic used.

2. In the Introduction line 84/85 the recent publication by Redenti et al. Nature November 14, 2024 should be added.

R: We agree that this is an interesting and important publication and have added this where suggested.

3. The time at which measurements were done should be mentioned for the culture as well as for the ex vivo analysis.

R: We have made sure that the time of culture for both bacterial and mammalian cell experiments have been explicitly stated and thank the reviewer for bringing this to our attention.

Editorial requests:

- We plan to publish your manuscript as Report, as there are only 5 main figures and 4 EV figures. For a Scientific Report we require that results and discussion sections are combined in a single chapter called "Results & Discussion". Please do this for your manuscript. For more details please refer to our guide to authors:

<http://www.embopress.org/page/journal/14693178/authorguide#researcharticleguide>

R: We have combined the results and discussion into a single paragraph.

- Please order the manuscript sections like this, using these names:

Title page - Abstract - Keywords - Introduction - Results & Discussion - Methods - Data availability section - Acknowledgements - Disclosure and Competing Interests Statement - References - Figure legends - Expanded View Figure legends

R: The manuscript now adheres to this order of sections.

- Please remove the part "ORCID for corresponding author" from the manuscript text file. The ORCID will be directly linked to the corresponding author in the online version of the paper.

R: We have removed this paragraph.

- Please make sure that all figure panels (main and EV figures) are called out separately and sequentially. Presently, there seem to be no separate callouts for panels 2G, 3B, 5A and 5F. Please check.

R: All panels are now specifically mentioned in the main text.

- Please check again that the number "n" for how many independent experiments were performed, their nature (biological versus technical replicates), the bars and error bars (e.g. SEM, SD) and the test used to calculate p-values is indicated in the respective figure legends (main and EV figures). Please also check that all the p-values are explained in the legend, and that these fit to those shown in the figure. Please provide statistical testing where applicable. Please avoid the phrase 'independent experiment', but clearly state if these were biological or technical replicates. Please also indicate (e.g. with n.s.) if testing was performed, but the differences are not significant. In case n=2, please show the data as separate datapoints without error bars and statistics. See also:

<http://www.embopress.org/page/journal/14693178/authorguide#statisticalanalysis>

R: All statistical testing with NS have been added where testing was done, Fig 1B where $n=2$ mean and SEM + statistics have been removed and data is shown as individual points. Exact p-values have been stated in the figure legend.

2B + 2D + 2F are all the same n described in the study design in 2A, seems to be redundant to call them out for each one. 2G has the actual counts on the axis, so also seems redundant to put it. 5B + 5D + 5F are all the same n called in 5A.

If $n < 5$, please show single datapoints for diagrams. Presently, it seems 'n.s.' is not indicated in several diagrams. Moreover:

R: NS added; datapoints shown where applicable in all figures.

- Please provide information related to n in the legends of figures 2B, D, F, G; 4A, B; 5B, D, F.

R: n has been added but see answer above for 2B + 2D + 2F and 5B + 5D + 5F.

- Please note that $n=2$ in figures 1B. Thus, please show the two datasets separately and remove the statistics.

R: This has been implemented.

- Please define the error bars in the legends of figures EV2 B-D.

R: This has been implemented.

- Please note that the figure EV2 subfigure "D" is mislabeled as figure "C" in the manuscript. This needs to be rectified.

R: This has been corrected.

- Please provide exact p values in the legends of figures 1B, D, E; 2B, C, G; 3C; 4A, B; 5B, C, G.

R: All p-values are now explicitly stated in the legend of the figures.

- Please move the information shown in Table 2 to the Reagents and Tools table and update the callout. Then please remove Table 2 from the manuscript text file. Moreover, please add callouts to the reagents and tools table in the methods section where applicable.

R: Has been moved and corrected in text callout.

- Please add scale bars of similar style and thickness to microscopic images, using clearly visible black or white bars (depending on the background). Please place these in the lower right corner of the images themselves. Please do not write on or near the bars in the image but define the size in the respective figure legend.

R: A 100 μ M scale-bar has been placed in the lower right corner and is described in the legend.

- Please use our reference format ('et al' needs to be used after 10 author names; DOIs should only be used for preprints and datasets that have not been published yet):

R: The reference format has been updated in the main text and bibliography.

- The should the light microscopic images (left column) and the IF images (right column) in Fig. EV3 show the same tumor image for each row? I seems for some images this is not the case. Please check.

R: The images are indeed the same tumor in 1 row but are from adjacent slides which gives rise to some differences in their appearance. We have updated the legend to clearly state that the images are on adjacent slides.

- For the source data for Fig. 3B please provide a TIFF file of the actual images shown, as suggested.

R: The TIFF files have been uploaded in the source data for figure 3 and re-uploaded.

In addition, I would need from you uploaded separately:

- a short, two-sentence summary of the manuscript (not more than 35 words).

R:

We engineered *E. coli* Nissle 1917 to produce indole-3-acetic acid (IAA), an AhR agonist. This strain modulates the tumor microenvironment, enhances T-cell infiltration, and suppresses tumor growth, demonstrating a promising microbiome-based cancer immunotherapy.

(Also uploaded separately)

- two to four short (!) bullet points highlighting the key findings of your study (two lines each).

R:

- **Engineered Microbiome Therapeutics:** *E. coli* Nissle 1917 was engineered to produce indole-3-acetic acid (IAA), effectively activating AhR without affecting bacterial fitness.

- **Tumor Immunomodulation:** IAA-producing bacteria increased CD4⁺/CD8⁺ T-cell infiltration and reduced IL-17A levels, promoting AhR-driven anti-tumor immunity.
- **Therapeutic Efficacy:** Treatment suppressed tumor growth, improved survival, and induced lasting immunity in rechallenged mice.

(Also uploaded separately)

- a schematic summary figure as separate file that provides a sketch of the major findings (not a data image) in jpeg or tiff format (with the exact width of 550 pixels and a height of not more than 400 pixels) that can be used as a visual synopsis on our website.

R: We have provided the graphical abstract in two formats:

1. A high-quality TIFF file to ensure optimal clarity and detail.
2. A PNG file resized to 550 pixels wide at 96 DPI, as specified in the submission guidelines.

While we have adhered to the requested dimensions for the PNG version, we feel that the resulting quality is suboptimal and would suggest using the higher-quality TIFF file if possible.

Morten Sommer
Technical University of Denmark
Novo Nordisk Foundation Center for Biosustainability
Kemitorvet 220
Kgs. Lyngby, Zealand 2800
Denmark

Dear Dr. Sommer,

I am very pleased to accept your manuscript for publication in the next available issue of EMBO reports. Thank you for your contribution to our journal.

Yours sincerely,
